# Fluvastatin suppresses breast cancer initiation and progression via targeting CYP4Z1
Huilong Li [1,5], Ying Chen[1,5], Wanjin Shi[1], Zheng Miao[1], Yu Lu[1], Xuedan Han[1], Haitao Chen[1], Yunnan Zhang[2], Miaomiao Niu[1], Shengtao Xu[1], Hai Qin [3] ✉, Lufeng Zheng [1] ✉ & Qianqian Guo [2,4] ✉

Breast cancer ranks highest globally in terms of both incidence and mortality rates among female malignancies. Elucidating the molecular mechanisms driving breast cancer initiation and progression, as well as identifying novel therapeutic agents, remains a critical unmet medical need. This study aimed to identify FDA-approved CYP4Z1 inhibitors with anti-breast cancer activity through a drug repurposing strategy, thereby providing preclinical evidence for potential clinical adjuvant therapies. Fluvastatin was identified as a concentration-dependent CYP4Z1 inhibitor through molecular docking and site-directed mutagenesis studies, binding to critical residues Lys109, Pro444, and Arg450 in the enzyme's active site. Functional studies demonstrated that Fluvastatin significantly attenuated cancer stem cell properties, migratory/invasive capacities, and epithelial-mesenchymal transition in breast cancer cell lines. In vivo experiments revealed that fluvastatin suppressed primary tumor growth and lung metastasis in xenograft models, while delaying mammary tumorigenesis in *PyMT-MMTV-CYP4Z1* transgenic mice. Notably, this effect was less pronounced in *PyMT-MMTV* wild-type controls. This study establishes Fluvastatin as a novel CYP4Z1-targeted therapeutic candidate for breast cancer, providing preclinical validation for its potential use in combination therapies.

Breast cancer ranks second globally among all malignancies while remaining the principal cause of cancer-related mortality in women worldwide, with metastatic progression and disease recurrence representing the primary determinants of adverse outcomes[1,2]. Breast cancer stem cells (BCSCs), representing a distinct subpopulation within the breast cancer cell hierarchy, possess unlimited proliferative potential and tumor-initiating ability and are the main reasons for chemotherapy resistance and tumor recurrence[3,4]. Emerging evidence indicates chemotherapy-driven transdifferentiation of non-stem cancer cells into BCSCs contributes to enhanced metastatic potential and disease relapse[5,6]. In the absence of reliable targets, there are currently no drugs target BCSCs, and it is rather challenging to target this cell population clinically. Therefore, it is of particular importance to discover new targets for this cell population and develop drugs based on these targets to target BCSCs.

Cytochrome P450 Family 4 Subfamily Z Member 1 (CYP4Z1) was initially discovered by Rieger when searching for genes that are specifically and highly expressed in breast cancer tissues[7]. Subsequent studies have found that the expression level of CYP4Z1 is positively correlated with the progression of breast cancer[8,9]. In addition, the concentration of CYP4Z1 antibodies in the serum of breast cancer patients is significantly higher than that in the serum of normal individuals[10,11]. Mounting evidence supports the association between CYP4Z1 overexpression in multiple cancer types and adverse clinical outcomes, rendering it a putative prognostic biomarker and promising target for oncotherapy[7,12,13]. It has been reported that CYP4Z1 is abnormally highly expressed on the surface of breast cancer cells but not expressed in healthy control breast cells[10]. Our previous research found that the expression level of CYP4Z1 in BCSCs tumorspheres were significantly upregulated. Knockdown of CYP4Z1 could attenuate the stemness of breast cancer cells by inhibiting the PI3K/Akt and ERK signaling pathways[14]. The overexpression of CYP4Z1 would promote tumor angiogenesis[15] and inhibit the apoptosis of breast cancer cells[16]. These results suggest that CYP4Z1 could be a strong candidate for breast cancer treatment via targeting BCSCs.

[1]School of Life Science and Technology, Department of Medicinal Chemistry, School of Pharmacy, China Pharmaceutical University, Nanjing, Jiangsu Province, China. [2]Department of Pharmacy, The Affiliated Cancer Hospital of Zhengzhou University & Henan Cancer Hospital, Zhengzhou, P. R. China. [3]Department of Clinical Laboratory, Beijing Jishuitan Hospital, Guizhou Hospital, Guiyang City, Guizhou Province, China. [4]State Key Laboratory of Neurology and Oncology Drug Development, Nanjing, China. [5]These authors contributed equally: Huilong Li, Ying Chen. ✉e-mail: 18786665889@163.com; zhlf@cpu.edu.cn; zlyygqq4265@zzu.edu.cn

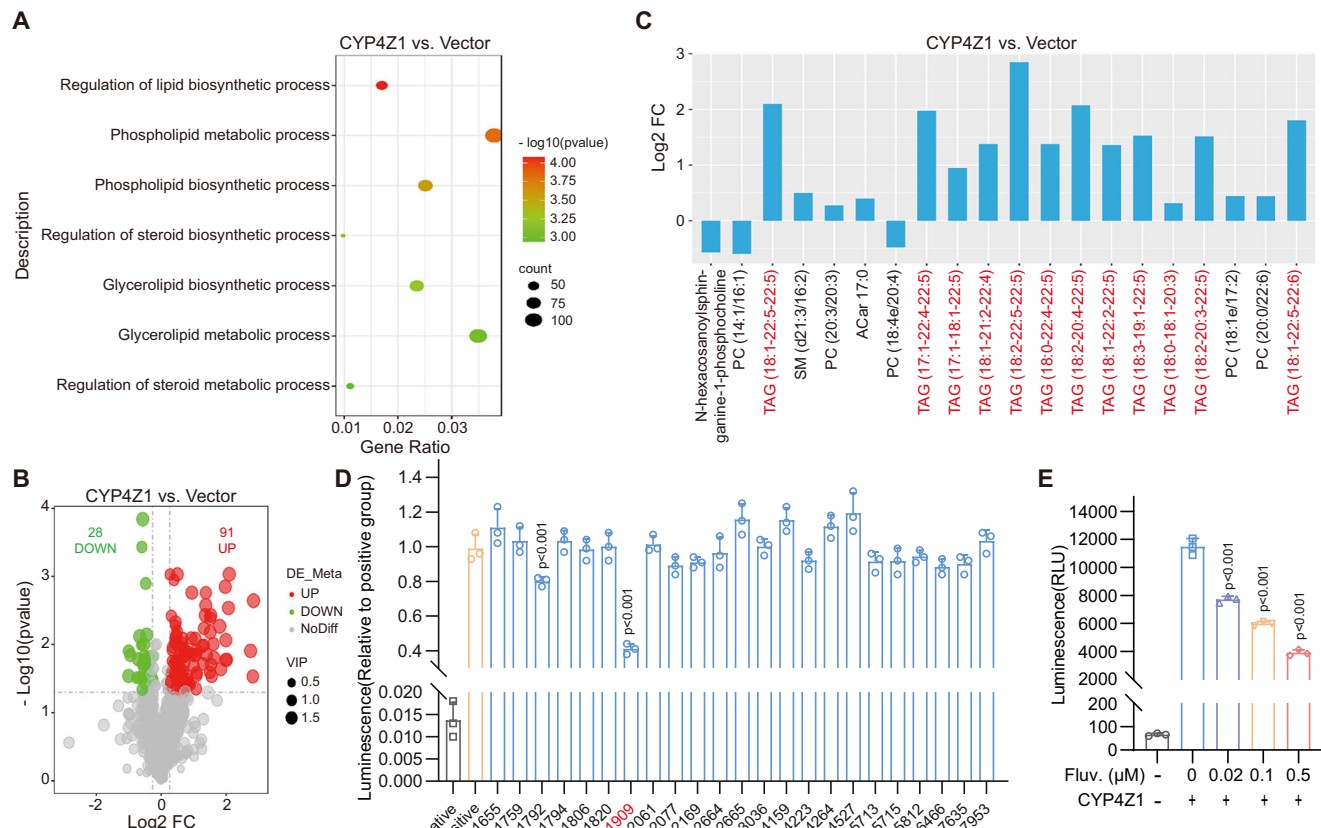

**Fig. 1 | CYP4Z1 overexpression enriches lipid metabolism and fluvastatin inhibits CYP4Z1 enzymatic activity. A** GO enrichment analysis of the biological process with CYP4Z1 overexpression in MDA-MB-231 cells. *P* value was calculated by *t* test and represents the level of significance. **B** Results of differential lipid compound analysis in MDA-MB-231 cells with CYP4Z1 overexpression. The screening of differential lipid compounds primarily referred to three parameters: variable importance in projection (VIP), fold change (FC), and *P* value. VIP denotes the variable importance in projection value of the first principal component of the partial least squares-discriminant analysis (PLS-DA) model, which reflects the contribution of lipid compounds to group separation; FC represents the fold change, defined as the ratio of the mean quantitation values of all biological replicates for each lipid compound between the comparison groups. The screening thresholds were set as follows: VIP > 1.0, FC > 1.2 or FC < 0.833, and *P* value < 0.05. **C** The 20 lipid compounds with the largest fold changes in lipidomics analysis. **D** Enzymatic assays were performed to screen a lipid-regulating drug library (detailed information is provided in Supplementary Table 1). The positive control group used a solvent (all drugs were dissolved in PBS). *n* = 3 biologically independent samples. **E** Enzymatic assays were conducted to determine the inhibitory activity of fluvastatin (dissolved in PBS) against CYP4Z1 at different concentrations. *n* = 3 biologically independent samples. Error bars in the figures represent the standard error of the mean (SEM).

Clinical studies have demonstrated that breast cancer patients exhibit abnormally elevated lipid levels, with a high incidence of new-onset dyslipidemia observed following chemotherapy[17–21]. In this study, we found that CYP4Z1 can significantly promote lipid anabolism and upregulate triglyceride (TAG) levels in breast cancer cells by using targeted lipid metabolomics technology. Based on the role of CYP4Z1 in lipid metabolism regulation and the characteristics of abnormal lipid metabolism in breast cancer patients undergoing chemotherapy, we hypothesize that lipid-lowering drugs could be screened for potential candidates targeting CYP4Z1, thereby providing a novel therapeutic approach for BCSC-targeted treatment in breast cancer. Although we have already discovered several CYP4Z1 inhibitors that can suppress the stem cell characteristics of breast cancer[22–24], these inhibitors cannot be used clinically. As an already marketed drug, fluvastatin has unique advantages. Drug repurposing is the process of developing new uses outside the original indications of old drugs[25,26]. Compared with the development of new drugs for specific indications, the drug repurposing strategy has several advantages; for example, a lower failure rate, less time required for drug development, and reduced financial costs[27]. The present study searched for potential drugs that target and inhibit CYP4Z1 for the treatment of breast cancer based on a small molecule library of lipid-lowering drugs containing 23 FDA-approved drugs. And we identified fluvastatin as a candidate drug for the treatment of breast cancer. Fluvastatin is an oral HMG-CoA reductase inhibitor commonly used for the

treatment of primary hypercholesterolemia and primary mixed dyslipidemia[28]. However, its potential role in cancer treatment is still unclear.

Here, we screened a compound library containing 23 FDA-approved lipid-lowering drugs. And found that fluvastatin can significantly suppresses breast cancer initiation and progression. Our mechanistic results suggest that Fluvastatin can significantly inhibit the enzymatic activity of CYP4Z1 by bounding to the Lys109, Pro444, and Arg450 sites, thereby repressing the stemness upregulation by CYP4Z1.

## Results

### CYP4Z1 overexpression enriches lipid metabolism and fluvastatin inhibits CYP4Z1 enzymatic activity

Our transcriptome analysis (GSE116984) revealed that the cell line with CYP4Z1 overexpression could significantly enrich the biological processes related to lipid metabolism (Fig. 1A). Then we further conducted targeted lipid metabolomics (10.6084/m9.figshare.30866477) on breast cancer cells with CYP4Z1 overexpression. As shown in Fig. 1B, compared with the control group, a total of 119 differential lipid metabolites were identified in the CYP4Z1 overexpression group, among which 91 were upregulated lipid metabolites and 28 were downregulated. And we noted that CYP4Z1 overexpression could remarkably upregulated the triglyceride (TAG) metabolism level (Fig. 1C). Given that the lipid levels of breast cancer patients are abnormally upregulated during treatment[19–22,24] and CYP4Z1

induces the upregulation of lipid metabolism in breast cancer cells, we screened potential anticancer drugs from a lipid-lowering compound library containing 23 FDA-approved drugs (Supplementary Table 1). The results showed that fluvastatin (S1909) could significantly inhibit the enzymatic activity of CYP4Z1 (Fig. 1D). Additionally, the results of enzymatic activity detection at multiple concentrations indicated that fluvastatin could inhibit the enzymatic activity of CYP4Z1 in a concentration-dependent manner (Fig. 1E). Specifically, we detected the impacts of fluvastatin and simvastatin(S1792) treatment on stemness of the MDA-MB-231 cell line. As presented in Supplementary Fig. 1A–C, fluvastatin showed a more potent inhibitory effect on breast cancer cell stemness than S1792.

The above results preliminarily suggest that fluvastatin can inhibit the enzymatic activity of CYP4Z1 and has a potential inhibitory effect on breast cancer, which will be further verified in the following experiments.

## Fluvastatin attenuates the stemness and migration of breast cancer cells in vitro

We initially assessed the cytotoxicity of fluvastatin with breast-related cells. Notably, the $IC_{50}$ of fluvastatin against the human mammary epithelial cell line (MCF-10A) was substantially higher than that against human and mouse breast cancer cell lines. This finding suggests that at the same dosage, fluvastatin exerts a cytotoxic effect on tumor cells while inducing relatively minimal damage to normal tissues (Fig. 2A and Supplementary Fig. 2A–D). In addition, we found that fluvastatin can enhance the sensitivity of breast cancer cells to adriamycin (Fig.2B). We investigated the effects of fluvastatin on the stemness and migration of breast cancer cells. Notably, our prior work and other published studies have demonstrated that MDA-MB-231 and MCF-7-Adr cells exhibit stronger stemness and migratory capacity compared to other breast cancer cell subtypes. Therefore, we selected MDA-MB-231 and MCF-7-Adr cells as the primary research models for this part of the study[14,29–31].

Immunoblotting analysis revealed that fluvastatin reduced the expression levels of stemness markers in MDA-MB-231 and MCF-7-Adr breast cancer cells, and this inhibitory effect exhibited a concentration-dependent pattern (Fig. 2C). Transwell assays demonstrated that fluvastatin suppressed the migratory and invasive abilities of the breast cancer cell in a concentration-dependent pattern (Fig. 2D–G). In vitro tumorsphere formation assays under non-adherent cult.conditions are used to quantify CSCs frequency in tumor cell populations[32]. The results revealed that fluvastatin could impede the tumorsphere formation ability in a concentration-dependent manner (Fig. 2H–K). Consistently, the fluvastatin treatment remarkably decreased the proportion of the CSCs cell subpopulation ($CD24^-CD44^+$)[33] (Fig. 2L–O).

Collectively, these findings confirm that fluvastatin can reduce the stemness and migratory potential of breast cancer cells—importantly, this effect is observed at a concentration that exerts no significant toxicological impact on normal mammary epithelial cells.

## Fluvastatin inhibits tumor growth and metastasis

To validate our in vitro findings, we further conducted in vivo experiments using *Balb/c-nu* nude mice. Specifically, we established subcutaneous xenograft tumor models in these nude mice and combined this with an extreme limiting dilution assay (ELDA) to assess tumor-related phenotypes (Fig. 3A). After subcutaneous xenograft tumors formed in *Balb/c-nu* nude mice, fluvastatin was administered at regular intervals of every 7 days. The results demonstrated that fluvastatin effectively suppressed the growth of xenograft tumors, as evidenced by reduced tumor volume and/or weight compared to the control group. Regrettably, there was no significant difference in the number of tumors formed across groups with different cell inoculation amounts. Due to the lack of variability in tumor formation frequency, the ELDA could not be performed as originally planned (Fig. 3B–D). We further performed IF analysis to assess the expression of the stemness marker ALDH1A1 and the proliferation marker Ki67 in the tumors. The results demonstrated significantly reduced ALDH1A1 and Ki67 expression in fluvastatin-treated tumors compared with vehicle controls (Fig. 3E–G).

To evaluate the effect of fluvastatin on breast cancer metastasis, we subsequently established a lung metastasis model (Fig. 3H). H&E staining was performed on lung tissues from the model mice. The results demonstrated that fluvastatin significantly alleviated breast cancer lung metastasis, as evidenced by a marked reduction in the number of lung metastatic nodules compared with the control group (Fig. 3I, J and Supplementary Fig. 3A). Importantly, there was no significant difference in body weight between mice administered fluvastatin and those in the control group (Fig. 3K), which further supports that fluvastatin exhibits relatively low in vivo toxicity under the experimental treatment conditions.

Collectively, the above findings indicate that fluvastatin exerts multifaceted inhibitory effects on breast cancer: it suppresses the growth, proliferation, and metastatic capacity of breast cancer cells, while also significantly reducing the stemness of breast cancer cells. These results collectively support fluvastatin as a potential candidate for targeted breast cancer intervention.

## Fluvastatin attenuates the stemness of breast cancer cells via targeting CYP4Z1

Our previous research demonstrated that the overexpression of CYP4Z1 activates the PI3K/AKT pathway in breast cancer[14]. Fluvastatin can inhibit the activation of the PI3K/AKT pathway in breast cancer cells and reverse the overactivation of the PI3K/AKT pathway caused by the overexpression of CYP4Z1 (Fig. 4A). To further verify the targeting of fluvastatin to CYP4Z1, we exogenously added TAG to mimic the results of CYP4Z1 overexpression (CYP4Z1 was not overexpressed). After exogenous addition of TAG, stemness markers, p-PI3K and p-AKT were upregulated, and the down regulation effect of fluvastatin administration on them was negligible (Fig. 4B and Supplementary Fig. 4A). Similarly, the ability of fluvastatin to reverse the enhanced migration and invasion abilities caused by TAG was weaker than that of the control group (Fig. 4C, D and Supplementary Fig. 4B, C). The results of in vitro tumorsphere formation assays and flow cytometry also showed that the effect of fluvastatin on reducing the proportion of stem cells was weakened or disappeared after the addition of TAG (Fig. 4E–G and Supplementary Fig. 4D).

Subsequently, we confirmed through rescue experiments that CYP4Z1 can rescue the downregulation of stemness markers (Fig. 4H), the weakening of migration and invasion abilities (Fig. 4I, J and Supplementary Fig. 4E, F), and the reduction in the proportion of stem cells (Fig.4K–M and Supplementary Fig. 4G) caused by fluvastatin. Furthermore, we established in vitro CYP4Z1 knockdown models. Both fluvastatin treatment and CYP4Z1 knockdown attenuated breast cancer cell stemness and inhibited PI3K/AKT pathway activation (Supplementary Fig. 5A). Consistent with these findings, both interventions reduced the proportion of cancer stem cells in breast cancer cells (Supplementary Fig. 5F, G). Additionally, Transwell assays further confirmed that these two treatments impaired the migration and invasion capabilities of breast cancer cells (Supplementary Fig. 5B–E). To investigate whether there is a difference in the efficacy of fluvastatin between CYP4Z1 knockdown and normal models, we compared the relative inhibition rates of the aforementioned indicators. The results showed that in the CYP4Z1 knockdown model, the relative inhibition rates of fluvastatin on most indicators were significantly lower than those in the normal model—this finding indicates that CYP4Z1 is a key target of fluvastatin (Supplementary Fig. 5H, I). However, fluvastatin still exerted a certain degree of efficacy in the CYP4Z1 knockdown model, which suggests that fluvastatin may act on additional targets beyond CYP4Z1 (Supplementary Fig. 5H, I).

We further demonstrated the targeting of fluvastatin to CYP4Z1 through A922500, a potent diacylglycerol acyltransferase 1 inhibitor that can effectively block TAG synthesis. We administered A922500 and fluvastatin to cells overexpressing CYP4Z1. A922500 and fluvastatin showed similar tumor suppressing abilities, and the ability of fluvastatin to weaken the stemness of breast cancer cells was attenuated when co-administered with A922500 (Fig. 4N–S and Supplementary Fig. 6A–C).

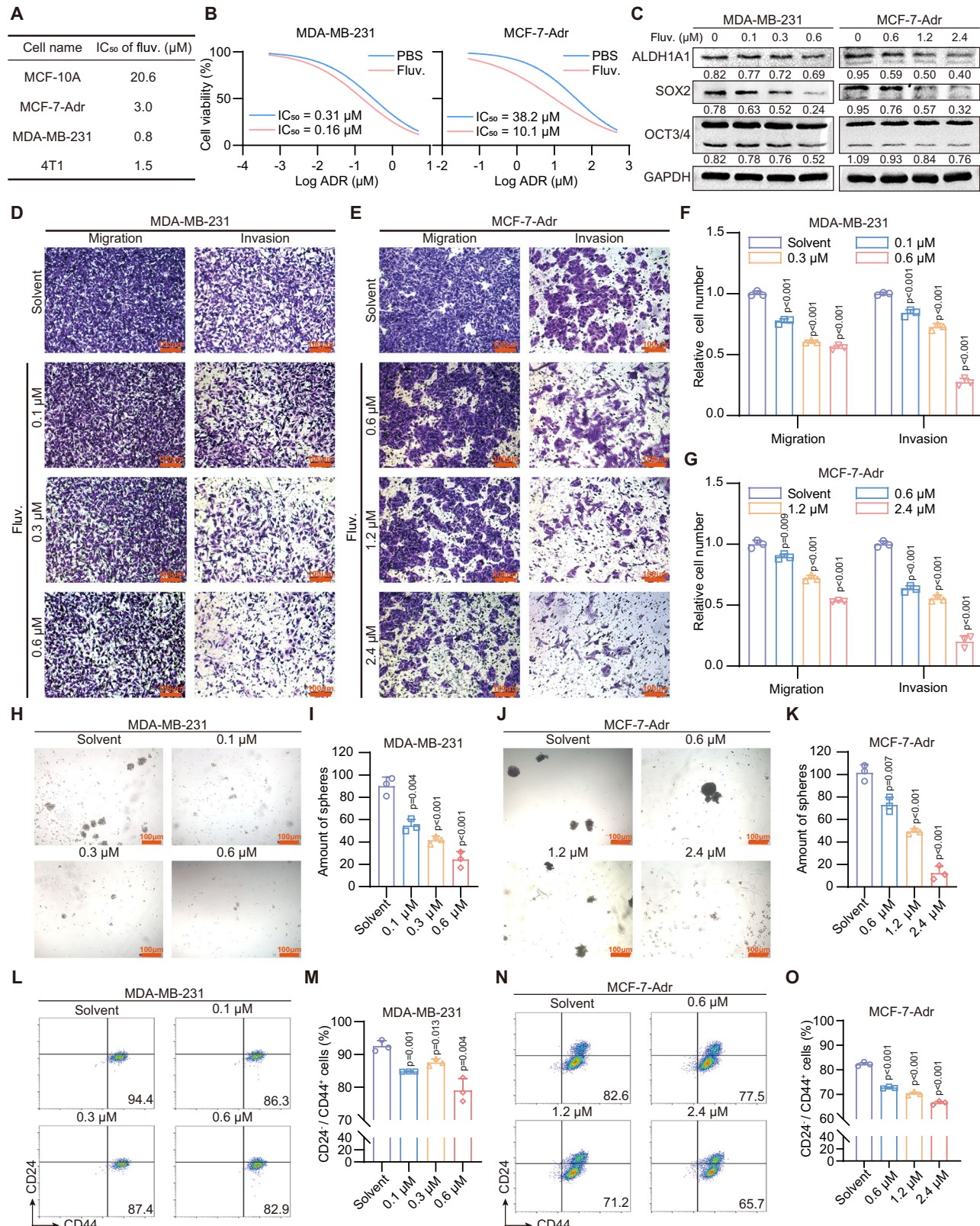

In conclusion, these experiments suggest that, on the one hand, CYP4Z1 enhances the stem cell characteristics of breast cancer cells by activating the PI3K/AKT pathway, and on the other hand, it enhances these characteristics by increasing the synthesis of TAG. Fluvastatin, however, can attenuate the stemness of breast cancer cells via targeting CYP4Z1.

**Fluvastatin attenuates the stemness of breast cancer cells via binding to the Lys109, Pro444, and Arg450 of CYP4Z1**

To confirm the binding of fluvastatin and CYP4Z1, we predicted the binding sites of fluvastatin and CYP4Z1 through 3D modeling combined with molecular docking. The binding sites were identified as Lys109, Pro444, and Arg450 (Fig. 5A). We constructed three single-point mutants of

**Fig. 2 | Fluvastatin attenuates the stemness and migration of breast cancer cells in vitro. A** The IC$_{50}$ of fluvastatin in each cell line was determined using CCK-8 assay. $n = 3$ biologically independent samples. **B** The IC$_{50}$ of adriamycin in MDA-MB-231 and MCF-7-Adr cells was determined using CCK-8 assay. $n = 3$ biologically independent samples. **C** Protein abundance of stemness markers; the numbers represent the ratio of each marker's abundance to that of GAPDH. **D–G** Representative images of Transwell assays in MDA-MB-231 and MCF-7-Adr cells. Statistical analysis is presented, with relative values calculated as the ratio of each value to the mean value of the solvent group. $n = 3$ biologically independent

samples. **H–K** Representative images of tumorsphere formation assays in MDA-MB-231 and MCF-7-Adr cells. Statistical analysis is shown, where each value represents the number of spheres in one well of a 24-well plate. $n = 3$ biologically independent samples. **L–O** Flow cytometric analysis of CD24 and CD44 expression in MDA-MB-231 and MCF-7-Adr cells. Each group included three biological replicates, and statistical analysis is provided. $n = 3$ biologically independent samples. Scale bar, 100 µm. Fluvastatin was dissolved in PBS. Error bars in the figures represent SEM.

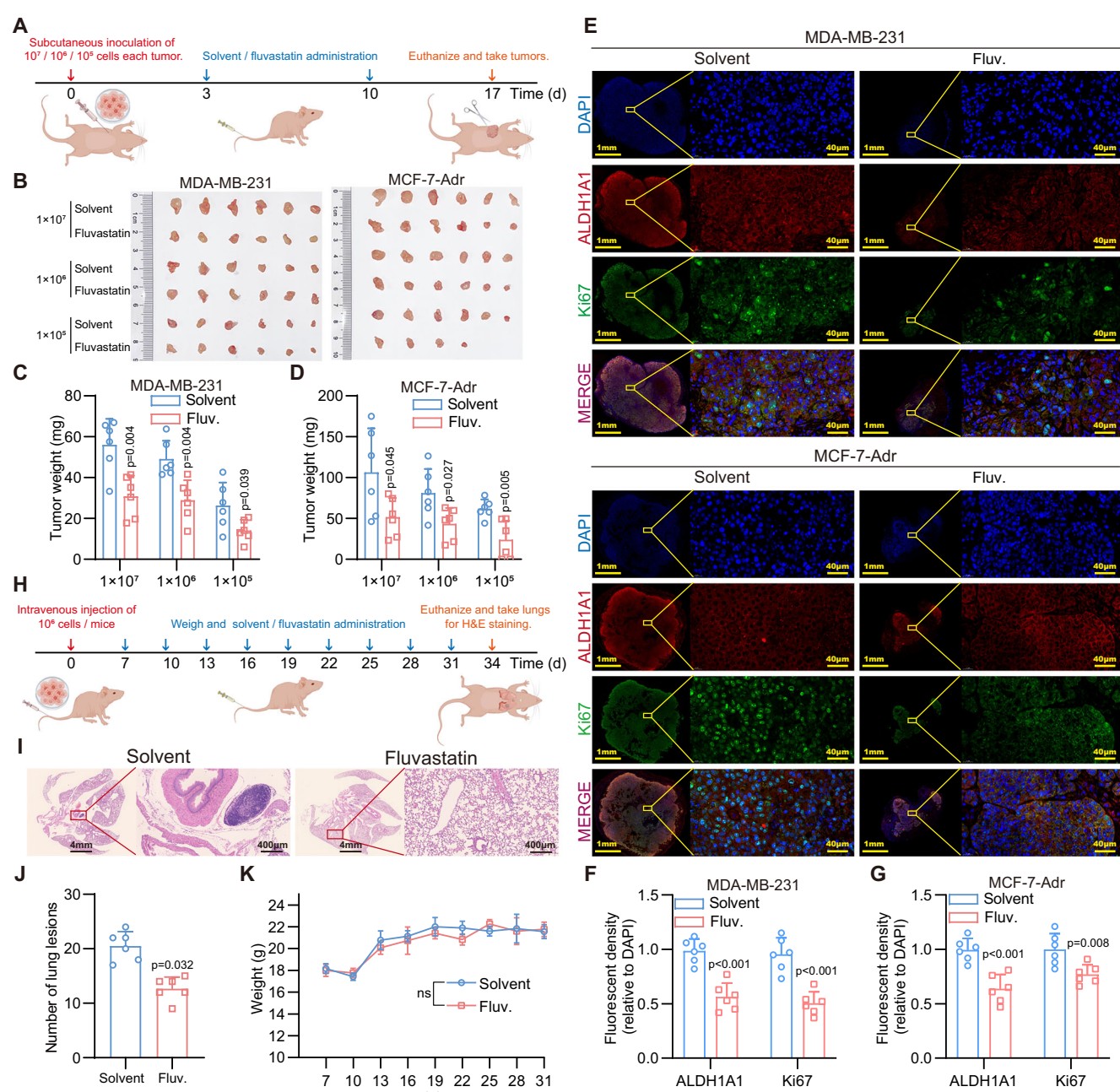

**Fig. 3 | Fluvastatin inhibits tumor growth and metastasis. A** Schematic diagram illustrating the construction and experimental design of the xenotransplantation model. **B–D** Effects of fluvastatin administration on the size and weight of MDA-MB-231 and MCF-7-Adr xenograft tumors. Fluvastatin was administered at a dose of 10 mg/kg and dissolved in PBS. $n = 6$ biologically independent animals. **E–G** Immunofluorescence analysis and quantification results of the stemness marker ALDH1A1 and proliferation marker Ki67 in xenograft tumors. Scale bars,

1 mm and 40 µm. $n = 6$ biologically independent animals. **H** The schematic diagram of the construction and experiment of the lung metastasis model. **I, J** The H&E staining results of the lungs with or without fluvastatin administration (10 mg/kg, dissolved in PBS) and the quantification of the number of lesions. Scale bars, 4 mm and 400 µm. $n = 6$ biologically independent animals. **K** Changes in body weight of nude mice during fluvastatin treatment. $n = 6$ biologically independent animals. Error bars in the figures represent SEM.

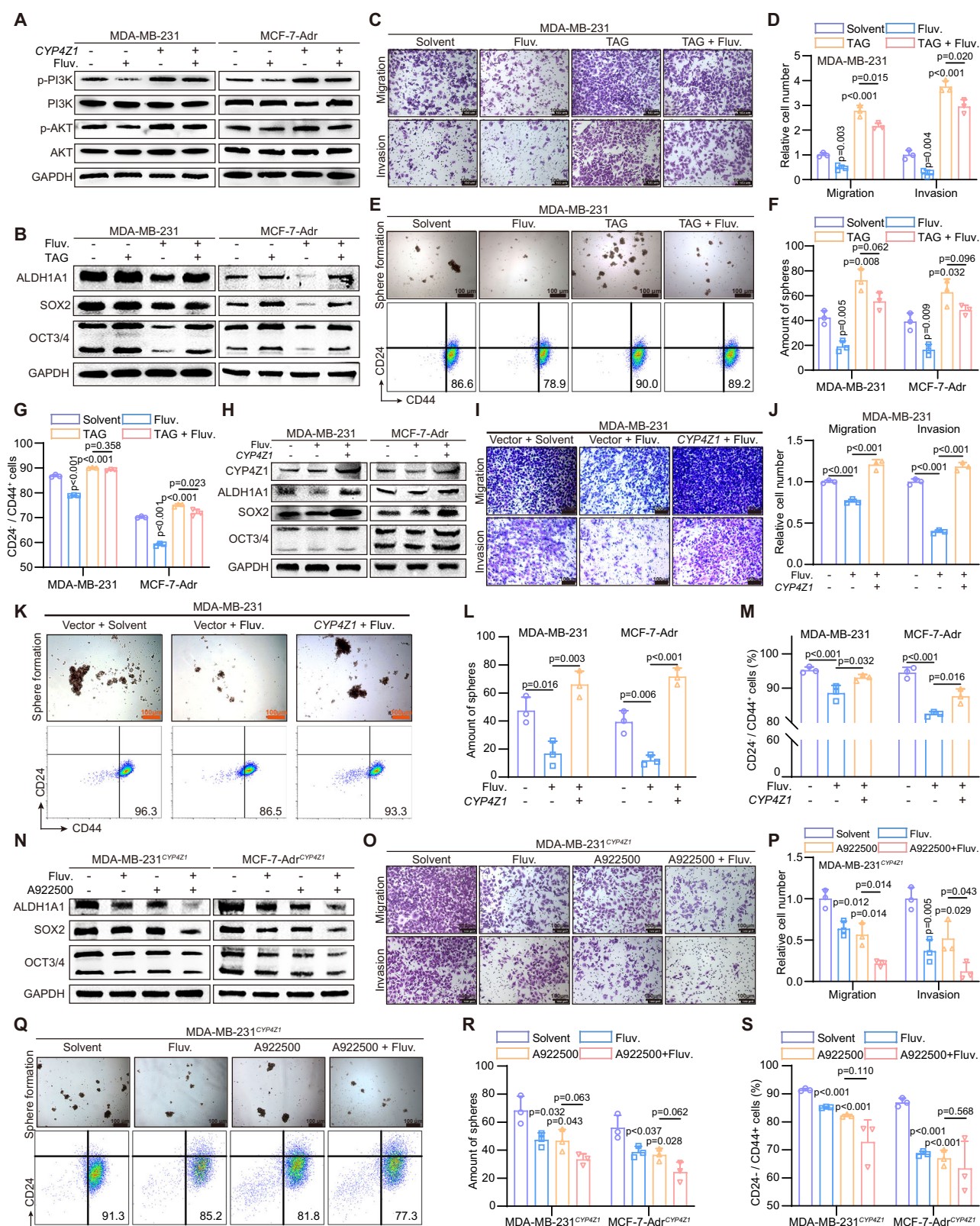

CYP4Z1, namely K109R, P444A, and R450M (Fig. 5B). Enzymatic activity assays were performed on the wild-type and mutant CYP4Z1 proteins. The results showed that, compared with the wild-type protein, the enzymatic activity of the mutants was decreased, and fluvastatin could no longer further reduce the enzymatic activity of the CYP4Z1 mutants (Fig. 5C, D). This

indicates that the binding site of fluvastatin and CYP4Z1 is located within its enzymatic active center domain.

To eliminate the influence of endogenous CYP4Z1, we selected the mouse breast cancer cell line 4T1, which does not express CYP4Z1, for in vitro efficacy verification. Both the wild-type CYP4Z1 and mutants were

**Fig. 4 | Fluvastatin attenuates the stemness of breast cancer cells via targeting CYP4Z1. A** Changes in protein abundance of PI3K, p-PI3K, AKT, and p-AKT in MDA-MB-231 and MCF-7-Adr cells following CYP4Z1 overexpression and fluvastatin treatment. **B** Protein abundance changes of stemness markers in MDA-MB-231 and MCF-7-Adr cells treated with fluvastatin (0.6 or 2.4 μM, dissolved in PBS) and TAG (300 μM, dissolved in DMSO). **C, D** Representative images of Transwell assays in MDA-MB-231 cells and corresponding statistical analysis. Relative values were calculated as the ratio of each value to the mean value of the solvent group. $n = 3$ biologically independent samples. **E** Representative images of tumorsphere formation assays in MDA-MB-231 cells; FACS analysis of CD24 and CD44 expression in MDA-MB-231 cells. $n = 3$ biologically independent samples. **F, G** Statistical analysis of the results from (**E**) and Supplementary Fig. S4D. Each group included three biological replicates. $n = 3$ biologically independent samples. **H** Changes in protein abundance of stemness markers in MDA-MB-231 and MCF-7-Adr cells following CYP4Z1 overexpression and fluvastatin treatment. **I, J** Representative images of Transwell assays in MDA-MB-231 cells and corresponding statistical analysis. Relative values were calculated as the ratio of each value to the mean value of the NC group. $n = 3$ biologically independent samples. **K** Representative images of tumorsphere formation assays in MDA-MB-231 cells; FACS analysis of CD24 and CD44 expression in MDA-MB-231 cells. $n = 3$ biologically independent samples. **L, M** Statistical analysis of the results from (**K**) and Supplementary Fig. S4G. Each group included three biological replicates. $n = 3$ biologically independent samples. **N** Protein abundance changes of stemness markers in MDA-MB-231 and MCF-7-Adr cells treated with fluvastatin (0.6 or 2.4 μM, dissolved in PBS) and A922500 (50 μM, dissolved in DMSO). **O, P** Representative images of Transwell assays in MDA-MB-231$^{CYP4Z1}$ cells and corresponding statistical analysis. Relative values were calculated as the ratio of each value to the mean value of the solvent group. $n = 3$ biologically independent samples. **Q** Representative images of tumorsphere formation assays in MDA-MB-231$^{CYP4Z1}$ cells; FACS analysis of CD24 and CD44 expression in MDA-MB-231$^{CYP4Z1}$ cells. $n = 3$ biologically independent samples. **R, S** Statistical analysis of the results from (**Q**) and Supplementary Fig. S5C. Each group included three biological replicates. $n = 3$ biologically independent samples. Scale bar, 100 μm. Error bars in the figures represent SEM.

found to enhance cell stemness and promote the activation of the PI3K/AKT pathway. However, a key distinction emerged in the magnitude of these effects: the extent of stemness enhancement and PI3K/AKT pathway activation induced by each of the three mutants was weaker than that induced by the wild-type CYP4Z1 (Fig. 5E and Supplementary Fig. 7A). Fluvastatin exhibited the best inhibitory effect on the wild-type CYP4Z1, while its inhibitory effect on the mutants was relatively weak (Fig. 5F). Consistently, the results of in vitro tumorsphere formation assays demonstrated that both the wild-type and mutants of CYP4Z1 increased the proportion of stem cell populations in 4T1. After fluvastatin administration, the upregulation of the stem cell proportion caused by the wild-type CYP4Z1 could be significantly reversed, while the enhancement caused by the mutants was only slightly downregulated without significant differences (Fig. 5G, H).

Furthermore, we confirmed the binding of fluvastatin and CYP4Z1 through DARTS and CETSA. The results showed that after treatment with fluvastatin, the enzymatic resistance and thermal stability of the wild-type CYP4Z1 was increased, while those of mutants did not change significantly (Fig. 5I–L).

These findings confirmed that fluvastatin binds to the enzymatic active center of CYP4Z1, inhibits its enzymatic activity, and attenuates the effect of CYP4Z1 on stemness.

## Fluvastatin inhibits CYP4Z1-induced tumor formation and metastasis based on transgenic mice model

Finally, we determined whether Fluvastatin exerted any deleterious effects on the self-renewal capacity and multilineage differentiation potential of normal hematopoietic stem cells (HSCs). As shown in Supplementary Fig. 8A, there was no significant difference in body weight between the administration group and the solvent group, indicating that fluvastatin exhibits relatively low toxicity in vivo. After 30 days of administration, peripheral blood was collected from the mice for a complete blood cell count. The results demonstrated that all differentiated lineage cells were within the reference range (Supplementary Fig. 8B). After euthanizing the mice, bone marrow was harvested to extract primary bone marrow cells. The results of flow cytometry analysis indicated that neither *PyMT-MMTV* nor the knock-in of *CYP4Z1* had a significant impact on the hematopoietic-related cell subpopulations. Moreover, continuous administration of fluvastatin exerted a minimal effect on the hematopoietic capacity of mice (Supplementary Fig. 8C). Herein, we verified the safety of fluvastatin administration and confirmed that *CYP4Z1* knock-in does not affect the development and hematopoietic capacity of mice.

As CYP4Z1 is not endogenously expressed in mice, to further investigate the effect of fluvastatin on primary breast cancer, we generated transgenic mice with targeted expression of *CYP4Z1* transgene in the breast tissue (Fig. 6A) and breed them with the *PyMT-MMTV* to obtain *PyMT-MMTV* mice with specific *CYP4Z1* knock-in breast tissue (*PyMT-MMTV-*

*CYP4Z1*) (Fig. 6B). Mice were treated with fluvastatin for 30 days from day 70 accompanied with the examination of the body weights (Fig. 6C). Given that the development of breast cancer in mice is associated with mammary luminal progenitor cells and stem cells, and considering that the CD24$^+$CD29$^{hi}$ population is known to be enriched for stem/progenitor cells while the CD24$^+$CD29$^{lo}$ population retains some progenitor properties, we performed primary isolation and sorting of mammary tissues. Flow cytometry analysis revealed that knock in *CYP4Z1* increased the proportion of the CD24$^+$CD29$^{lo}$ cell subset in mammary epithelial cells of *PyMT-MMTV* mice, with no effect of fluvastatin administration on this subset. Meanwhile, knock in *CYP4Z1* also elevated the proportion of the CD24$^+$CD29$^{hi}$ subset, and fluvastatin treatment significantly reduced the proportion of this specific subset (Fig. 6D, E). Additionally, both mammary hyperplastic lesions and mammary tumors were increased in *PyMT-MMTV* and *PyMT-MMTV-CYP4Z1* mice, which was attenuated by fluvastatin administration (Fig. 6F, G). Notably, compared with *MMTV-PyMT* and knock in *CYP4Z1* significantly increased the number of metastatic lesions in the liver and lungs, this effect was also refined by fluvastatin administration (Fig. 6H–K). Therefore, our results confirmed that knock in *CYP4Z1* elevates the risks of *PyMT*-induced tumor formation and metastasis, which can be refined by the administration of fluvastatin.

Notably, even though fluvastatin exhibited superior efficacy in models expressing CYP4Z1 compared to those not expressing CYP4Z1, both the CYP4Z1 knockdown models and the *PyMT-MMTV* model still responded to fluvastatin (Supplementary Figs. 5H,I and 8D)—this suggests that fluvastatin possesses additional anti-tumor targets beyond CYP4Z1. Our results demonstrated that CYP4Z1 overexpression enhanced lipid metabolism and increased TAG levels in breast cancer cells (Fig. 1B, C). This observation raises the questions: given that lipid peroxidation is a critical driver of ferroptosis, does CYP4Z1 associate with ferroptosis? And could the activity of fluvastatin in regulating lipid metabolism be one of the mechanisms underlying its induction of ferroptosis in breast cancer cells? To address this question, we measured the intracellular GSH content, the results showed that neither CYP4Z1 overexpression nor fluvastatin treatment had an significant impact on GSH levels (Supplementary Fig. 9A). Additionally, we examined the expression levels of GPX4 and ACSL4. Regrettably, no significant changes in the expression of these two markers were observed across experimental groups (Supplementary Fig. 9B). Consistently, Prussian blue DAB-enhanced staining analysis revealed that fluvastatin treatment did not alter iron accumulation within the tumor (Supplementary Fig. 9C). Collectively, these results consistently indicate that fluvastatin does not induce ferroptosis in breast cancer cells.

## Discussion

CYP4Z1 was originally discovered by Rieger during an investigation aimed at finding genes that are both specific to and highly expressed in breast

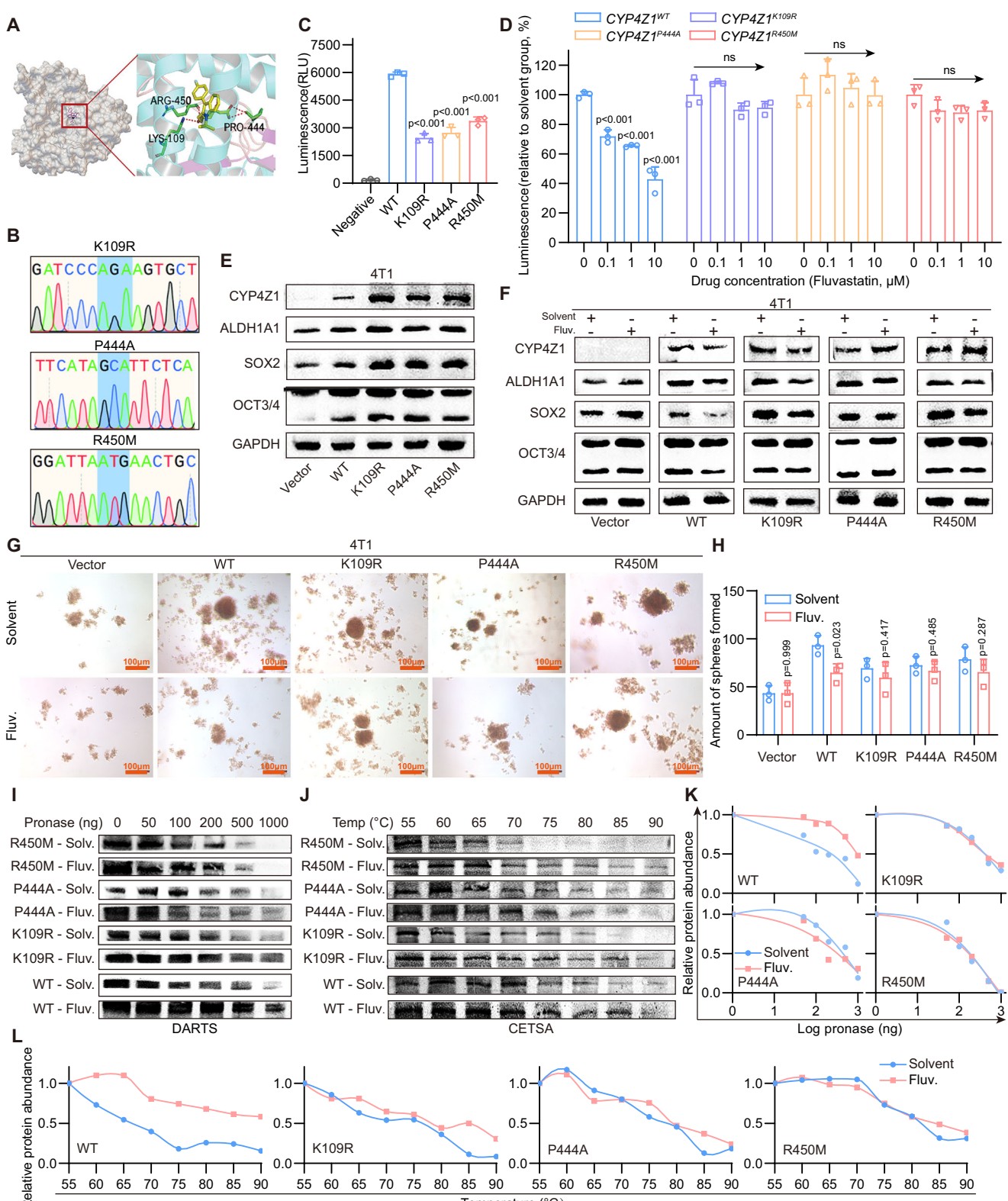

**Fig. 5 | Fluvastatin attenuates the stemness of breast cancer cells via binding to the Lys109, Pro444, and Arg450 of CYP4Z1. A** Amino acid residues involved in the binding between CYP4Z1 and fluvastatin, predicted via molecular docking. **B** Schematic diagrams illustrating point mutations at the three predicted binding sites. **C** Enzymatic assays to determine the enzymatic activity of CYP4Z1 and its mutants. $n = 3$ biologically independent samples. **D** Enzymatic assays to measure the relative enzymatic activity of CYP4Z1 and its mutants following treatment with fluvastatin at different concentrations. Relative values were calculated as the ratio of each value to the mean value of the solvent control group. $n = 3$ biologically independent samples. **E** Changes in protein abundance of

stemness markers in 4T1 cells after overexpression of CYP4Z1 and its mutants. **F** Changes in protein abundance of stemness markers in 4T1 cells overexpressing CYP4Z1 or its mutants, following treatment with fluvastatin (1.2 μM, dissolved in PBS).
**G, H** Representative images of tumorsphere formation assays in 4T1 cells and corresponding statistical analysis. Each value represents the number of spheres in one well of a 24-well plate. $n = 3$ biologically independent samples. **I–L** Results of drug affinity responsive target stability (DARTS) and cellular thermal shift assay (CETSA) for CYP4Z1 and its mutants, with corresponding quantitative analysis. Scale bar, 100 μm. Error bars in the figures represent SEM.

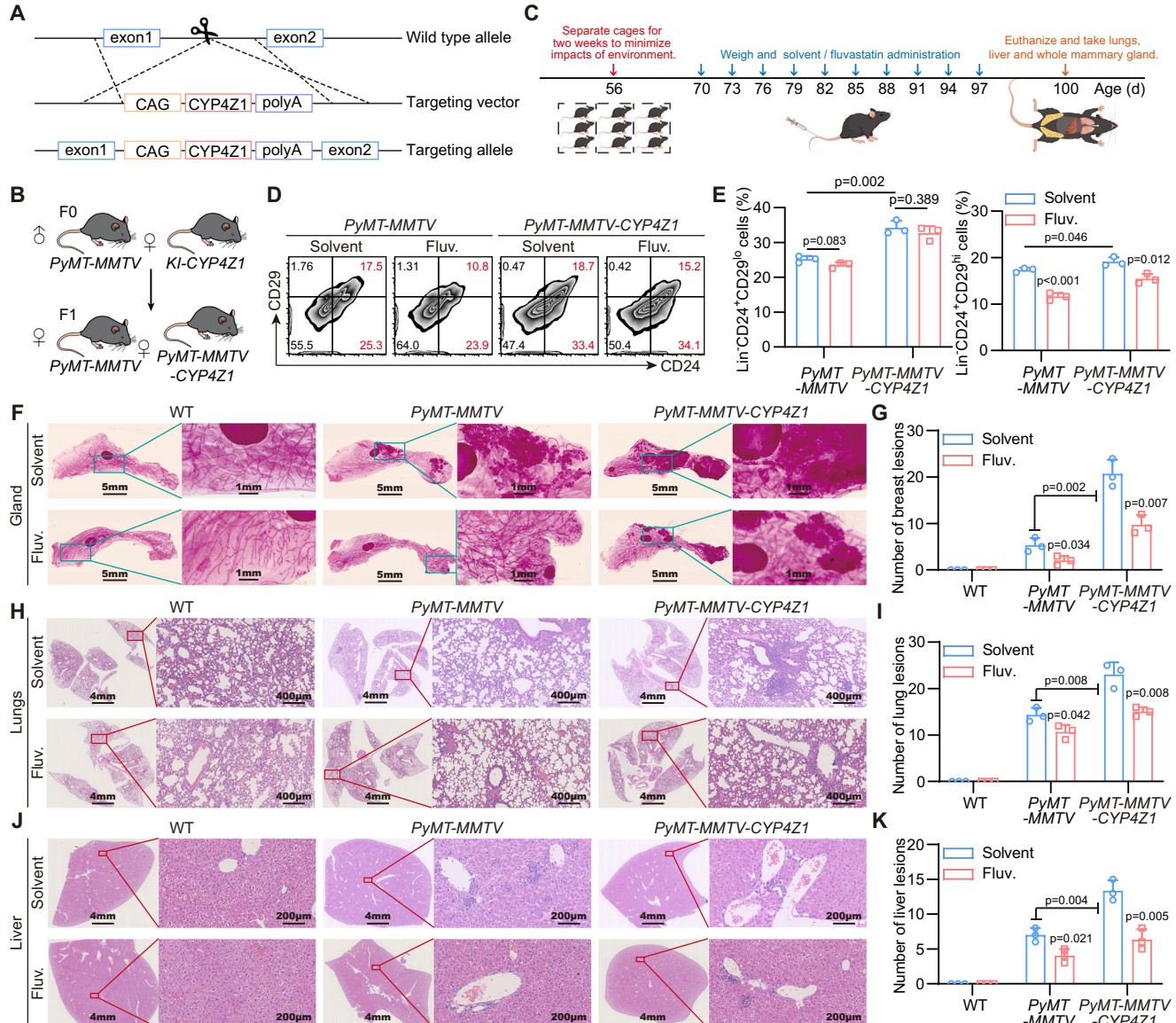

**Fig. 6 | Fluvastatin inhibits CYP4Z1-induced tumor formation and metastasis based on transgenic mice model. A, B** Schematic diagrams illustrating the construction and breeding of *CYP4Z1* knock-in transgenic mice. **C** Schematic diagram showing the construction and experimental design of the transgenic mouse model. Fluvastatin was administered at a dose of 10 mg/kg and dissolved in PBS. **D, E** FACS analysis of CD24 and CD29 expression in sorted mammary epithelial cells (Lin-), with corresponding statistical analysis. *n* = 3 biologically independent animals. **F, G** The whole mammary gland staining images and quantification of mammary gland lesions in transgenic mice after continuous treatment with fluvastatin or solvent. Scale bar, 5 mm and 1 mm. *n* = 3 biologically independent animals. **H, I** The H&E staining images of lung sections and quantification of lung lesions in transgenic mice after continuous treatment with fluvastatin or solvent. Scale bar, 4 mm and 400 μm. *n* = 3 biologically independent animals. **J, K** The H&E staining images of liver sections and quantification of liver lesions in transgenic mice after continuous treatment with fluvastatin or solvent. Scale bar, 4 mm and 200 μm. *n* = 3 biologically independent animals. Error bars in the figures represent SEM.

cancer tissues[7].Our prior research demonstrated that CYP4Z1 levels was significantly upregulated in breast cancer stem cell tumor microspheres, and the knockdown of CYP4Z1 was shown to diminish the stemness of breast cancer cells by inhibiting PI3K/AKT signaling pathway, thereby reversing the doxorubicin-resistant in these cells[14]. Consistent with our research, it was found that breast-specific knock-in of CYP4Z1 could not induce breast cancer, but could significantly upregulate the expression level of ERα, which is required for the proliferation of breast cancer-initiating cells (luminal type progenitors)[34,35]. Thus, CYP4Z1 is a key driver protein in promoting the stemness of breast cancer cells. In this study, employing drug repurposing strategies, we found that the lipid-lowering drug Fluvastatin effectively inhibits the enzyme activity of CYP4Z1, thereby significantly suppressing the initiation and progression of breast cancer. Further mechanistic research indicated that Fluvastatin exerts its inhibitory effect by bounding to the

Lys109, Pro444, and Arg450 sites on CYP4Z1, consequently down-regulating the stemness associated with CYP4Z1 activity.

In recent years, extensive research has been conducted on small-molecule inhibitors targeting the CYP4Z1 protein. Our research has also identified several CYP4Z1 inhibitors capable of suppressing breast cancer stemness; However, these inhibitors present numerous challenges. For instance, the in vivo toxicity and pharmacokinetic data of 1-Benzylimidazole remain uncharacterized[36], HET0016 exhibits poor selectivity[22], and the selectivity and pharmacological activity of some inhibitors are yet to be determined. Drug repurposing, as a promising approach in drug development, offers several unparalleled advantages, including shorter development timelines, reduced risk of failure, and lower costs[27]. There are numerous successful examples of drug repurposing, such as thalidomide and disulfiram[37-39]. Since some breast cancer patients

experience an upregulated dyslipidemia during treatment and the over-expression of CYP4Z1 can lead to intracellular lipid accumulation, we searched for potential drugs for breast cancer treatment among lipid-lowering drugs. We found that fluvastatin is a potent inhibitor of CYP4Z1 and can inhibit breast cancer both in vitro and in vivo. Fluvastatin is an oral HMG-CoA reductase inhibitor commonly used in the treatment of primary hypercholesterolemia and primary mixed dyslipidemia. It has been recently reported that the combination of fluvastatin and cisplatin can trigger endoplasmic reticulum stress and thus inhibit non-small cell lung cancer[40]. In pancreatic cancer, fluvastatin radiosensitizes radiotherapy-resistant cell lines by inhibiting the radiation-induced DNA damage repair response[41]. Another study pointed out that fluvastatin prevents lung metastasis in triple-negative breast cancer by triggering autophagy via the RhoB/PI3K/mTOR pathway[42]. These studies suggest that fluvastatin shows significant promise in the field of clinical oncology treatment. Our study suggests that fluvastatin may serve as a promising adjuvant therapy for breast cancer. Through a series of experiments, we confirmed that fluvastatin targets CYP4Z1 to suppress BCSCs. In a transgenic *PyMT-MMTV* mouse model, *CYP4Z1* knock-in significantly increased the risk of tumor formation and metastasis. Fluvastatin administration effectively inhibited these processes without adversely affecting the development or hematopoietic capacity of the mice, preliminarily supporting its safety for clinical application.

During the research process, we found that the expression efficiency of the *CYP4Z1* plasmid in 4T1 cells was increased after mutation, and the reason requires further exploration. In our screening, simvastatin (S1792) was also found to exhibit inhibitory activity against CYP4Z1; regrettably, its pharmacological effect was less potent than that of fluvastatin (Supplementary Fig. 1A–C). Another lipid-lowering drug, lovastatin (S2061), has been reported to induce the re-expression of human epidermal growth factor receptor 2 in triple-negative breast cancer, thereby making it sensitive to receptor tyrosine kinase inhibitors[43], but it did not exhibit inhibitory activity against CYP4Z1. We also discovered that CYP4Z1 knockdown models and *PyMT-MMTV* mice responded to fluvastatin (Fig. 6, Supplementary Fig. 5H, I). These results suggest that fluvastatin has multiple roles in the occurrence and development of breast cancer, and there may be other potential targets (excluding ferroptosis, Supplementary Fig. 9), which deserve further investigation. Nevertheless, in both in vitro and in vivo models, the efficacy of fluvastatin in models expressing CYP4Z1 was significantly superior to that in models not expressing CYP4Z1 (Supplementary Figs. 5H, I and 8D). Therefore, there is no denying that CYP4Z1 is one of the key targets through which fluvastatin exerts its anti-tumor effects.

In conclusion, our study provides evidence that fluvastatin may be a promising new anti-breast cancer drug. Its ability to inhibit CYP4Z1 enzymatic activity and suppress breast cancer progression both in vitro and in vivo offers new possibilities for the clinical adjuvant treatment of breast cancer. Future research should focus on optimizing the combination of fluvastatin with other therapies and further exploring its mechanism of action to fully realize its potential in breast cancer treatment.

## Methods
### Cell culture and reagents
Human breast cancer cell lines, including MCF-7-Adr cells (doxorubicin-resistant, exhibit stronger stemness and migratory capacity compared to MCF-7), and MDA-MB-231 cells were purchased from Cellverse (Shanghai, China), human embryonic kidney HEK-293T cells, human mammary epithelial MCF-10A cells, and murine breast cancer 4T1 cells were purchased from Procell (Wuhan, China). HEK-293T and MDA-MB-231 cells were cultivated in DMEM medium, while MCF-7-Adr and 4T1 cells were cultured in RPMI-1640 medium supplemented with 10% fetal bovine serum (OM625656, FBS, OmnimAbs, Shanghai, China). To sustain drug resistance, MCF-7-Adr was cultured with 1 μM adriamycin (HY-121309, MCE, Monmouth Junction, NJ, USA) in the medium. The marketed drug compound library for hypolipidemic indications was obtained from the Selleck active compound library. The specific drug concentration was determined based on the $IC_{50}$ of fluvastatin for the cell lines. Cells were treated with all

drugs for 48 h in vitro. The solvent of fluvastatin was PBS, and the solvent of TAG and A922500 was DMSO.

### Plasmid, si-*CYP4Z1* and cell transfection
The CYP4Z1 coding sequences were amplified via polymerase chain reaction (PCR) technology from a human cDNA template. Next, it was ligated into the pCDNA4 HisMaxB vector and verified by DNA sequencing. si-*CYP4Z1* sequence: -CAUUACCUUUCCAGAUGGATTdTdT- The transfection reagent used was Jet-PRIME (101000046, Polyplus, Illkirch, France), following the manufacturer's recommended protocol. The transfected cells were cultured in complete medium for 48 h to ensure adequate expression of the CYP4Z1 gene. The overexpression and knockdown methods for other genes were performed in the same manner. Primers sequences for *CYP4Z1* single-point mutation were included in Supplementary Table 2.

### RNA sequencing and data analysis
RNA sequencing and data analysis were conducted by Novogene (Beijing, China). Paired-end read sequences were aligned to the human reference genome (version mm10) using the default settings in STAR (version 2.6.1b) and quantified by HTSeq (version 0.11.0) in "intersection-strict" mode. Significant DEGs were identified as those with a false discovery rate (FDR) value above the threshold ($Q < 0.05$) and fold-change >2 using edgeR software (v3.2.0). The data are available in the Gene Expression Omnibus (GEO) database as GSE116984.

### Lipidomics identification and analysis
Lipidomics analysis was carried out by Beijing Nuohe Biotechnology Co. Ltd. The experimental protocol encompassed sample collection, lipid extraction, and LC-MS/MS detection. Mass spectrometry raw data were analyzed with Compound Discoverer 3.1 software for spectral analysis and database search to perform lipid profiling. Stringent quality control protocols were integrated to validate data integrity. Multivariate analysis, including PCA and PLS-DA, identified metabolic differences between groups. HCA and correlation analysis evaluated sample relationships and variable correlations.

### Enzyme activity assay for CYP4Z1
Pyrolytic P450-dependent bioluminescence assays were performed using Promega's protocol (Madison, WI, USA). CYP4Z1 enzymatic activity was evaluated using CYP4Z1 protein solution. Reaction mixtures containing 12 μL protein solution, 0.5 μL fluorescein-CEE (V8752, Promega, Madison, WI, USA), and inhibitor cocktail were preincubated at 37 °C for 10 min in 96-well white plates. Catalysis was initiated by adding 25 μL of 2× NADPH regeneration system (V9510, Promega, Madison, WI, USA). Following 30 min biotransformation at 37 °C, equal volumes of fluorescein detection reagent were added, and plates were vortexed prior to 20 min room temperature incubation. Luminescence intensity was measured using a microplate reader under dark conditions.

### Cell viability assay
Cell viability was then assessed using a CCK-8 cell proliferation assay kit (BMU106, Abbkine Scientific Co., Ltd). Cells were seeded at a density of 5000 cells per well in a total volume of 100 μL culture medium. After 24 h of incubation, the medium was replaced with drug-containing medium, and the cells were further cultured for 48 h. Subsequently, the medium was substituted with fresh medium supplemented with 10% CCK-8 solution. The cells were continuously incubated in a cell culture incubator for 1 h, and the absorbance was measured at a wavelength of 450 nm using a microplate reader. Relative cell viability was calculated based on the measured absorbance values.

### Western blotting and immunofluorescence (IF) analysis
Cell lysates were prepared with RIPA buffer (Beyotime, Beijing, China). Subsequently, these lysates were subjected to SDS-PAGE and transferred onto PVDF membranes (Merck Millipore, Billerica, MA, USA).

Membranes were blocked in 5% non-fat milk for 2 h at room temperature. After primary antibody incubation overnight at 4 °C, followed by species-specific HRP-conjugated secondary antibodies for 40 min at RT. Detection used High-sig ECL substrate (180-5001, Tanon, Shanghai, China) via enhanced chemiluminescence. ALDH1A1[44], OCT4[45], and SOX2[46] represent three typical markers of tumor stem cell characteristics. Tissues were fixed in 4% PFA for 48 h, paraffin-embedded, and sectioned. FFPE sections underwent dewaxing, hydration, and microwave-induced antigen retrieval in citrate buffer. Primary antibodies were incubated overnight at 4 °C, followed by Alexa Fluor-conjugated secondary antibodies for 1 h at RT. Slides were mounted with DAPI-containing antifade medium and imaged by confocal microscopy. All of related information of antibodies was mentioned in Supplementary Table 3.

### Transwell assays

In the chamber with or without Matrigel (Corning, NY, USA), an environment simulating cell invasion or migration was established[47]. The experiment was completed in transwell plate (Jet Biofil, Guangzhou, China). Serum-free medium and drugs were added to the chamber, while medium containing 20% FBS in cultured plate. A total of $10^5$ cells were seeded into the chamber and cultured for 48 h (MCF-7-Adr) or 24 h (MDA-MB-231). After fixing the cells with paraformaldehyde and staining them with crystal violet, chamber-embedded non-migratory and non-invasive cells were mechanically removed with a cotton-tipped. Subsequently, images were captured, and the number of migrated or invaded cells was counted.

### In vitro tumorsphere formation assays

Cells are cultivated under serum-free and non-adherent conditions to enrich the cancer stem/progenitor cell population, since only cancer stem/progenitor cells can endure and multiply within such an environment. In the DMEM/F12 cultured medium, MammoCult™ Proliferation Supplement (StemCell Technologies, Vancouver, BC, Canada) and fluvastatin were incorporated. 3000 cells were seeded in an ultralow-attachment plate. Images were taken and the number of mammospheres was counted 7 days later.

### Flow cytometry

Single cells were isolated from cell lines or primary mouse tissues and then incubated with corresponding species-specific antibodies at 4 °C for 30 min. For the analysis of stem cell populations, antibodies against CD24-PE, CD44-APC, and CD29-APC were utilized. Bone marrow stem cell analysis utilized Gr-1-FITC, Sca1-PE, B220-FITC, CD34-PE/Cyanine7, IgM-FITC, CD127-FITC, CD3ε-FITC, Ter119-FITC, c-Kit-APC, CD19-FITC, CD48-PerCP/Cyanine5.5, CD150-Brilliant Violet 421™, and CD16/32-Brilliant Violet 510™ antibodies. Flow cytometry data were acquired on a BD FACSCanto II flow cytometer (BD Biosciences, San Jose, CA, USA). All of related information of antibodies was mentioned in Supplementary Table 3.

The gating strategy was established as follows: BCSCs were identified as CD24⁻ CD44⁺ cells by gating on CD24 and CD44 expression profiles; Mammary gland luminal progenitor cells were defined as CD24⁺ CD29ˡᵒʷ cells, mammary gland luminal stem/progenitor cells were defined as CD24⁺ CD29ʰⁱᵍʰ cells by gating on CD24 and CD29 expression profiles. LSK cells were gated by first selecting Lin⁻ cells, followed by identifying Sca1⁺ cKit⁺ double-positive cells; MP cells were isolated from Lin⁻ cells by gating on cKit⁺ Sca1⁻ cell populations; Within the MP subset, CMPs were defined as Lin⁻ cKit⁺ Sca1⁻ CD16⁻ CD34⁺ cells, EMPs as Lin⁻ cKit⁺ Sca1⁻ CD16⁻ CD34⁻ cells, and GMPs as Lin⁻ cKit⁺ Sca1⁻ CD16⁺ CD34⁺ cells through sequential gating on CD16 and CD34 expression; HSCs were further purified from the LSK subset as Lin⁻ cKit⁺ Sca1⁺ CD48⁻ CD150⁺ cells by additional gating on CD48⁻ and CD150⁺ cells. All gating was performed using FlowJo software (Version 10.8.1).

### H&E staining analysis

The tissue paraffin block is sectioned and then placed in xylene for 15 min. The xylene is replaced and the step is repeated. Subsequently, the sections are successively treated with a series of ethanol solutions with decreasing concentrations and finally immersed in distilled water. Hematoxylin staining is carried out for 15 min. After rinsing, differentiation is performed in 1% hydrochloric acid ethanol, followed by blueing in an alkaline solution. Then, eosin staining is carried out for 5 min. Finally, dehydration is achieved through a series of ethanol solutions and xylene, and then the sections are mounted with neutral balsam.

### Molecular docking studies

For the docking process, the homology model of CYP4Z1, which was constructed in our previous research[24], served as the protein receptor. Protein preparation was performed using Schrodinger's Protein Preparation Wizard (v2022-3), including hydrogen addition, loop refinement, and restrained minimization (OPLS4 force field, RMSD ≤ 0.3 Å). Ligands were preprocessed with LigPrep (v3.5), generating all tautomeric and ionization states at pH 7.4 ± 0.5. A 3D grid box ($8 \times 8 \times 8$ Å³) was centered at the heme iron atom (coordinates: $x = 12.3$, $y = 45.7$, $z = 28.9$ in the homology model) using the Glide Grid Generation module. Flexible docking was executed in GlideSP (Standard Precision) mode (v7.8), allowing ligand torsion flexibility while keeping the receptor rigid. Pose filtering retained complexes with Glide Emodel energy ≤ −5 kcal/mol (Coulomb + vdW terms). Default parameters were used for all other docking settings, including the OPLS4 scoring function and water model. Final poses were visualized and analyzed in PyMOL (v2.5.4), with 2D interaction diagrams generated for key ligand-receptor contacts.

### Cellular thermal shift assay (CETSA)

CETSA is an experiment to detect the binding efficiency of drugs and target proteins in cells, based on the principle that target proteins usually become more stable when bound to drug molecules[48]. Collect HEK-293T cells overexpressing CYP4Z1 and extract proteins. Divide the obtained samples equally into two parts, and then incubate them with an equal volume of DMSO or fluvastatin (500 µM) at room temperature for 30 min. Subsequently, divide each group of samples into 8 portions and incubate them at a series of temperatures ranging from 55 to 90 °C, increasing the temperature by 5 °C each time, for 5 min. After the incubation period, cool the samples to room temperature, then centrifuge and collect the supernatant. Next, add 5× loading buffer to the supernatant and boil the mixture. Finally, load the samples onto a 10% sodium dodecyl sulfate-polyacrylamide gel (SDS-PAGE) for Western blotting analysis.

### Drug affinity responsive target stability (DARTS)

Cells were collected and proteins were extracted. The protein concentration was adjusted to 4–6 µg/µL using 10×TNC (Tris-NaCl-Casein) buffer. Then, the samples were divided into two equal parts. Each part was incubated with an equal volume of DMSO or fluvastatin (500 µM) at room temperature for 30 min. Subsequently, each group of samples was further divided into six portions and incubated with Pronase solution (10165921001, Roche, Basel, Switzerland) at different ratios at room temperature for 30 min. The reaction was terminated by adding 5× loading buffer and boiling the mixture. The resulting samples were subjected to 10% SDS-PAGE for western blotting.

### Isolation of lineage-negative neoplastic mammary epithelial cells (Lin--NECs) and pre-neoplastic mammary epithelial cells (pNECs)

Pre-neoplastic ductal structures and primary mammary tumors were microdissected into 1–2 mm³ fragments, with adipose tissues meticulously removed. Tissue fragments were enzymatically dissociated in a digestion cocktail. Following enzymatic digestion, cell suspensions were centrifuged at $300 \times g$ for 5 min at 4 °C. Pellets were sequentially treated with 0.25% trypsin-EDTA (25200056, Gibco, Grand Island, NY, USA) for 2 min, followed by 5 mg/mL Dispase II (17105041, Thermo Fisher Scientific, Waltham, MA, USA) plus 0.1 mg/mL DNase I (D5025, Sigma, St. Louis, MO, USA) for 5 min, and 0.64% NH₄Cl erythrocyte lysing buffer (A600611, Sangon Biotech, Shanghai, China) for 5 min, all at 37 °C. Digestion was quenched with complete DMEM/F12 medium containing 10% FBS. Single-cell

suspensions were filtered through a 40-μm cell strainer (352340, Corning, Corning, NY, USA) and washed twice with HBSS buffer (14025096, Gibco, Grand Island, NY, USA) supplemented with 0.5% BSA (A9647, Sigma, St. Louis, MO, USA). Lin⁻-NECs and pNECs were enriched using the EasySep™ Mouse Epithelial Cell Enrichment Kit II (19868, StemCell Technologies, Vancouver, BC, Canada) according to the manufacturer's protocol, with magnetic bead separation performed on a EasySep™ Magnet (18001, StemCell Technologies, Vancouver, BC, Canada).

#### Whole-mount staining of mammary gland
Fourth inguinal mammary glands were dissected and fixed in Carnoy's solution. Hydrated tissues were stained with carmine red, flattened, dehydrated, and mounted for microscopy.

#### Intracellular glutathione (GSH) detection assay
Collect cells ($5 \times 10^5$), wash twice with PBS to remove extracellular GSH. Lyse cells if needed, then centrifuge lysates at high speed for 10 min. Add 50 μL cell lysate or GSH standard to a microcentrifuge tube, followed by 150 μL 10% TCA. Vortex, incubate on ice for 10 min, then centrifuge at high speed for 10 min. Prepare 0.4 M EDTA solution. Transfer 50 μL supernatant to a new tube, add 200 μL 0.4 M EDTA and 25 μL 10 mM DTNB. Vortex, incubate at room temperature for 10 min. Measured absorbance at 412 nm with a spectrophotometer (transfer 100 μL mixture to a 96-well plate for analysis if needed). Calculate GSH concentration using a GSH standard curve (Abbkine, KTB1600-48T, China).

#### Prussian blue DAB-enhanced staining
Prussian blue DAB-enhanced staining was performed by Shanghai Ruchuang Biological Technology Co., Ltd., following the protocol: Paraffin blocks were sectioned, and the resulting sections were baked at 60℃. Subsequently, sections were dewaxed and dehydrated using xylene and absolute ethanol. Perls A and Perls B solutions were mixed at a 1:1 ratio, and sections were incubated in this mixture for 30 min for Prussian blue staining. Afterward, DAB chromogenic solution was dropped onto the sections for 2 min to develop color. Sections were then stained with hematoxylin for 30 s. Finally, the stained sections were dehydrated and mounted with neutral balsam.

#### Ethics approval and consent to participate
Animal studies were registered and conducted in strict compliance with guidelines and protocols submitted to and approved by the Animal Welfare and Ethics Committee (AWEC) of China Pharmaceutical University (approval number: 2021-10-017). We have complied with all relevant ethical regulations for animal use.

#### Animal feeding and management
Mice were group-housed at a density of 3 animals per cage in a standardized vivarium with environmental enrichment. Housing conditions were strictly controlled: temperature ($25 \pm 1$ ℃), relative humidity (40–75%), and a 12-h light/dark cycle (lights on at 07:00, off at 19:00). After purchase, all mice were allowed a 1-week acclimatization period to adapt to the vivarium environment before experimental procedures. Animals had ad libitum access to autoclaved rodent chow (standard formula) and continuous access to reverse osmosis-purified water. Daily monitoring was performed to assess general health (appetite, activity, mental status), and no adverse events were observed during acclimatization or the experimental period. Humane endpoints were predefined: mice would be humanely euthanized (administered sodium pentobarbital via intraperitoneal injection at a dose of 100 mg/kg body weight) if tumor diameter exceeded 1.5 cm, body weight decreased by >20% of the initial weight, or signs of severe distress (e.g., lethargy, inability to eat/drink) were observed.

#### Xenotransplantation
Female nude mice (strain: *BALB/c-nu*, substrain: nu/nu) aged 3 weeks (body weight: 12–15 g) were purchased from Huachuang Sino (Nanjing, China). All mice were confirmed to be pathogen-free, with no history of previous experimental procedures or underlying diseases. The animals were genetically intact (non-genetically modified) and maintained under specific pathogen-free (SPF) conditions throughout the study. MDA-MB-231 and MCF-7-Adr cells at densities of $10^7$, $10^6$, and $10^5$ were subcutaneously inoculated into the nude mice ($n = 6$ biologically independent animals, a total of 36 mice), respectively. Solvent (PBS)/fluvastatin (10 mg/kg) was administered weekly. Two weeks later, the nude mice were euthanized, and the tumors were harvested, weighed, and photographed.

#### In vivo metastasis study
Female nude mice (strain: *BALB/c-nu*, substrain: nu/nu) aged 3 weeks (body weight: 12–15 g) were purchased from Huachuang Sino (Nanjing, China). All mice were confirmed to be pathogen-free, with no history of previous experimental procedures or underlying diseases. The animals were genetically intact (non-genetically modified) and maintained under SPF conditions throughout the study. One week after injecting $10^6$ MDA-MB-231 cells into the tail vein of nude mice ($n = 6$ biologically independent animals, a total of 12 mice), solvent (PBS)/fluvastatin (10 mg/kg) was administered via the tail vein every 3 days, and the body weight of the nude mice was monitored. Three weeks later, the nude mice were euthanized, and the lungs were harvested for H&E staining analysis.

#### Transgenic mouse strains and tumor monitoring
Two transgenic mouse strains on a *C57BL/6 J* background were used: *CYP4Z1*-transgenic mice and *PyMT-MMTV* transgenic mice (stock number: 022974, Shanghai Nanfang Model Biotechnology, China) expressing the polyomavirus middle T antigen (*PyMT*) under the control of the mouse mammary tumor virus (*MMTV*) promoter. Female littermate wild-type *C57BL/6 J* mice were used as controls for all experiments. Mice were 4 weeks old at weaning (body weight: 18–22 g), pathogen-free, and genetically verified for the presence of the target transgenes (*CYP4Z1* or *PyMT-MMTV*) via PCR before the experiment. No previous experimental procedures were performed on these animals. For spontaneous tumorigenesis studies ($n = 3$ biologically independent animals, a total of 18 mice), 4-week-old post-weaning female mice were subjected to biweekly abdominal palpation starting at 8 weeks of age to detect breast tumor formation. Pulmonary metastases were evaluated via macroscopic examination of lung and liver tissues collected 21 days after initial tumor detection. Investigators remained unblinded to genotype-based group allocations throughout experimental procedures and outcome assessments. Animals were assigned to cohorts based solely on genetic background without randomization. Sample size was determined empirically without formal statistical power calculations. All procedures conformed to protocols approved by the Shanghai Nanfang Model Biotechnology Co., Ltd. Institutional Animal Care and Use Committee (IACUC; approval number: SNFM-2019-0026, Shanghai, China).

#### Statistics and reproducibility
Statistical analysis was performed using Prism 10.1.2 (GraphPad Software, La Jolla, CA, USA). Error bars in the figures represent the standard error of the mean (SEM). For two-group comparisons, two-tailed unpaired t-tests were carried out. For multiple group comparisons, a two-way ANOVA of post hoc test was implemented. $P$ values less than 0.05 were considered statistically significant. Individual values are reported in Supplementary Data.

A minimum of three biological replicates were employed, based on preliminary data from the lab, to ensure statistical robustness for the assays conducted. Blinding was not implemented during the analysis of the results as there was no specific preconception about the expected outcomes.

#### Reporting summary
Further information on research design is available in the Nature Portfolio Reporting Summary linked to this article.

#### Data availability
The RNA-seq data used in this study were previously generated and published in our prior work[14]. The raw data have been deposited in the GEO

database under accession codes GSE116984. In this study, these data were reanalyzed to identify biological metabolism associated with CYP4Z1. The lipidomics data have been deposited in the Figshare database under accession https://doi.org/10.6084/m9.figshare.30866477. The numerical source data for the graphs is found in Supplementary Data. Unedited western blots is found in Supplementary Fig. 10–13. All data are presented within the article and supplementary online data. All other data are available from the corresponding author on reasonable request.

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

## Acknowledgements

This work was supported by the National Natural Science Foundation of China (Grant No. 82204432, 82473955, 82173842), Henan Province Science and Technology Research Project(No.252102311191), Henan provincial Medical Science and Technology Research Project (No.SBGJ202502035). The Opening Foundation of State Key Laboratory of Neurology and Oncology Drug Development. WU JIEPING Medical Foundation (No. 320.6750.2023-05-7), Guizhou Provincial Basic Research Program(Natural Science) (Qian Ke He Ji Chu-[2024] Youth 020), 2025 Hospital-Level Scientific Research Fund of Beiiing Jishuitan Hospital Guizhou Hospital (JGYYK[2025]02), the Fundamental Research Funds for the Central Universities (2632025TD04), and the Priority Academic Program Development (PAPD) of Jiangsu Higher Education Institutions. The graphical abstract was drawn by Figdraw.

## Author contributions

Lufeng Zheng, Qianqian Guo and Hai Qin designed the research. Huilong Li and Ying Chen analyzed the data. Huilong Li, Ying Chen, Wanjin Shi, Zheng Miao, Yu Lu, Xuedan Han, Haitao Chen, Yunnan Zhang, Miaomiao Niu and Shengtao Xu performed the research. Huilong Li and Ying Chen wrote the paper. Lufeng Zheng, Qianqian Guo and Hai Qin reviewed this paper. All data were generated in-house, and no paper mill was used. All authors agree to be accountable for all aspects of work ensuring integrity and accuracy.

## Competing interests

The authors declare no competing interests.
