## [Transparent Peer Review file · Communications Biology]

Fluvastatin suppresses breast cancer initiation and progression via targeting CYP4Z1

Corresponding Author: Professor Lufeng Zheng

Version 0:

Reviewer comments:

Reviewer #1

(Remarks to the Author)

While previous studies have reported the effects of the fluvastatin on breast cancer proliferation and metastasis, this manuscript by Li and Chen et al offers novelty by elucidating the mechanistic aspects through the involvement of CYP4Z1. The study is of potential interest and the conclusions of the study are supported by the experiments. However, some major concerns need to be addressed that are essential to support the main conclusions of the study:

Major Points

1. Figure 1 is based on bioinformatic analyses. While it is valuable to combine in silico results with in vitro and in vivo data, these analyses should be explained in detail in both the Methods and Results sections, including the filters applied. Moreover, there appears to be another compound showing significant results (S1792). How do the authors explain why they did not include this compound in the manuscript? What exactly are those 23 FDA-approved lipid-regulating drugs? Providing an explanation or citing an appropriate reference would be useful.
2. The design of the experiments in Figure 3 is confusing, with several pieces of information either missing or inconsistent.
 - Is it necessary to use three different cell numbers for MDA-MB-231 cells? This is a well-established cell line for in vivo experiments.
 - Why was fluvastatin administered weekly in the tumor growth assay in Figure 3B, but every three days in Figure 3H?
 - What was the in vivo dose of fluvastatin?
 - Why is the experimental window for fluvastatin administration so short? To assess potential toxicity, the drug should be administered for a longer duration (e.g., 4 weeks for primary tumor models).
 - There is no body weight graph for the subcutaneous tumor model.
 - In Figure 3F, are the tumors from the 1×10^7 cell group or from another group?
 - Why are there only three tumors in each group in Figure 3F, while Figure 3C shows six tumors in the graph?Overall, the results presented in this figure are not convincing and require clarification.
3. Figure 4 could be strengthened by assessing changes in the PI3K/Akt pathway after exogenous addition of TAG, which would provide complementary evidence supporting the findings.
4. Some parts of the manuscript give the impression of being drafted quickly, and a more careful revision could greatly improve clarity and readability.

Minor Points

1. There are typos in the graphical abstract: durgs and bresat.
2. From lines 86 to 95, there are several grammatical errors that should be corrected.
3. Overall, the figure legends are weak and do not provide clear explanations.
4. Please specify the duration of in vitro treatment with fluvastatin (i.e., how many days the cells were treated).
5. Please clearly indicate the solvent used for fluvastatin, either in the main text or in the Methods section.

Reviewer #2

(Remarks to the Author)

In this study, the authors investigated the anti-tumor effects of fluvastatin, conducting both in vitro and in vivo experiments to show that fluvastatin may inhibit cancer stem cell properties by directly binding to and blocking the action of CYP4Z1.

Overall, the presented data indicate that fluvastatin has anti-tumor activity; however, it remains uncertain how much of this effect is attributable specifically to CYP4Z1. The following major and minor concerns need to be addressed:

1. In both the in vitro and in vivo models, only CYP4Z1 overexpression systems were used. Because CYP4Z1 overexpression represents an artificial condition, knockdown or knockout models are needed to demonstrate that CYP4Z1 is critical for the anti-tumor effects of fluvastatin.
2. In Figure 4A, CYP4Z1 overexpression does not appear to markedly affect PI3K/AKT phosphorylation, contrary to the authors' claim.
3. In Figure 5, the results appear contradictory. On one hand, all of the mutants exhibited reduced enzymatic activity (Fig. 5C); on the other hand, overexpression of these mutants led to increased stem cell marker expression (Fig. 5E). It remains unclear whether these mutants also activate downstream signaling pathways, such as PI3K/AKT.
4. In Figures 6D–E, the quality of the FACS data is poor. Furthermore, while the CD24⁺CD29hi population is known to be enriched for stem/progenitor cells, the CD24⁺CD29lo population also retains some progenitor cell properties. Overexpression of CYP4Z1 is expected to expand the CD24⁺CD29hi population. In addition, previous publications report that the CD24⁺CD29lo population accounts for approximately 20–30% of mammary epithelial cells.
5. The legends are too short and lacks some details.

Reviewer #3

(Remarks to the Author)

This is a well-designed and comprehensive study that investigates the potential of the FDA-approved lipid-lowering drug, fluvastatin, as a therapeutic agent for breast cancer by targeting the enzyme, CYP4Z1. The paper provides a strong body of evidence, using a combination of in vitro and in vivo models, to support its central hypothesis.

Key findings:

1. CYP4Z1 and Lipid Metabolism: Overexpression of CYP4Z1 enhances triglyceride (TAG) accumulation in breast cancer, consistent with clinical dyslipidemia.
2. Drug Repurposing: Screening identified fluvastatin as a potent inhibitor of CYP4Z1 enzymatic activity.
3. In Vitro Effects: Fluvastatin selectively reduced breast cancer cell viability, stemness (ALDH1A1, SOX2, OCT3/4, CD24⁻/CD44⁺), and migration, while sparing normal cells. It also enhanced adriamycin sensitivity.
4. In Vivo Effects: In xenograft and lung metastasis models, fluvastatin suppressed tumor growth, metastasis, and stemness markers with low toxicity.
5. Mechanism: Fluvastatin binds CYP4Z1 at Lys109, Pro444, and Arg450, inhibits PI3K/AKT signaling, and its effects are reversed by TAG or CYP4Z1 overexpression.
6. Transgenic Models: In CYP4Z1 knock-in mice, fluvastatin reduced PyMT-driven tumor growth and metastasis without impairing hematopoiesis, supporting clinical potential.

Strengths:

1. Comprehensive approach spanning drug repurposing, mechanistic studies, xenograft, and transgenic models.
2. Strong use of CYP4Z1 transgenic mice for in vivo relevance.
3. Rigorous mechanistic validation with docking, mutagenesis, CETSA, and DARTS.
4. Clinically relevant, addressing dyslipidemia in breast cancer patients.

Weaknesses:

1. Transgenic mouse experiments lacked randomization and power calculations, limiting generalizability.
2. Fluvastatin also affected PyMT-MMTV mice without CYP4Z1, suggesting off-target effects needing clarification.
3. Further studies should assess combination strategies to optimize clinical application.

Question:

1. Your study shows fluvastatin suppresses breast cancer stem cells via CYP4Z1. However, PyMT-MMTV control mice without CYP4Z1 also responded, implying additional mechanisms. What other targets or pathways might explain this effect? Have you considered its role as an HMG-CoA reductase inhibitor and potential CYP4Z1-independent impact on breast cancer?

2. CYP4Z1 overexpression elevates lipid metabolism and TAG levels in breast cancer cells, linking it to lipid peroxidation—the key driver of ferroptosis. Could fluvastatin's inhibitory effect on CYP4Z1 and lipid metabolism be a mechanism for inducing ferroptosis in breast cancer cells? Have you considered measuring markers of ferroptosis, such as lipid peroxidation (e.g., malondialdehyde or 4-hydroxynonenal) or the expression levels of key regulators like GPX4 and ACSL4, in your in vitro or in vivo models? Exploring this pathway would provide a more complete picture of how fluvastatin's modulation of lipid metabolism leads to tumor suppression.

Version 1:

Reviewer comments:

Reviewer #2

(Remarks to the Author)

In the revised manuscript, the authors have adequately addressed the questions raised by the reviewers.

Reviewer #3

(Remarks to the Author)

My primary questions centered on the potential for off-target effects of fluvastatin and the possibility of ferroptosis induction. The authors have provided new data for both points.

For my first question, the authors conducted in vitro experiments in CYP4Z1 knockdown cells, which validated the presence of a CYP4Z1-independent mechanism. They have now clearly articulated a dual mechanism of action for fluvastatin, incorporating both its direct inhibition of CYP4Z1 and its HMG-CoA reductase inhibitory activity. This detailed explanation significantly strengthens the mechanistic discussion of the paper.

For my second question, the authors conducted a thorough investigation into whether fluvastatin induces ferroptosis. Although the results were negative, they provided clear evidence by measuring key markers such as glutathione (GSH) levels, GPX4/ACSL4 expression, and intratumoral iron deposition. This conclusively rules out ferroptosis as a mechanism and demonstrates a commitment to scientific rigor.

The authors have addressed my concerns with diligence and have improved the overall quality of the manuscript. I believe the paper is now in good shape for publication.

Sincerely,

Dear Dr. Christina Karlsson Rosenthal and Kaliya Georgieva

We are very grateful to receive your interim decision on our manuscript (COMMSBIO-25-4861) entitled “**Fluvastatin suppresses breast cancer initiation and progression via targeting CYP4Z1**”. We are now submitting our revised manuscript for your consideration of publication in **Communications Biology**.

Thank you for your insightful comments and suggestions. We would like to take this opportunity to appreciate the careful evaluation and expert opinions on our manuscript by you and the three reviewers. Each query has then been cautiously considered and responded to with experiments and the corresponding parts in the manuscript flagged with yellow background. The details are listed below point by point according to the editor’s and reviewer’s comments.

Reviewer #1

1. Figure 1 is based on bioinformatic analyses. While it is valuable to combine in silico results with in vitro and in vivo data, these analyses should be explained in detail in both the Methods and Results sections, including the filters applied.

Response: Thanks for your valuable advice. Sincerely grateful to the reviewers for pointing out our negligence, we supplemented the RNA-seq-related analysis methods and filters applied in “**Materials and methods**” section and “**Results**” section.

The revised parts are shown below (red colored in the text). The corresponding passages are flagged with yellow background in the manuscript.

In “Materials and methods” section

RNA sequencing and data analysis

RNA sequencing and data analysis were conducted by Novogene (Beijing, China). Paired-end read sequences were aligned to the human reference genome (version mm10) using the default settings in STAR (version 2.6.1b) and quantified by HTSeq (version 0.11.0) in “intersection-strict” mode. Significant DEGs were identified as those with a false discovery rate (FDR) value above the threshold ($Q < 0.05$) and fold-change >2 using edgeR software (v3.2.0). The data are available in the Gene Expression Omnibus (GEO) database as GSE116984.

In “Results” section

Figure 1. CYP4Z1 overexpression enriches lipid metabolism and fluvastatin inhibits CYP4Z1 enzymatic activity.

(A) GO enrichment analysis of the biological process with CYP4Z1 overexpression in MDA-MB-231 cells. P value was calculated by t-test and represents the level of significance

(B) Results of differential lipid compound analysis in MDA-MB-231 cells with CYP4Z1 overexpression. The screening of differential lipid compounds primarily referred to three parameters: variable importance in projection (VIP), fold change (FC), and P-value. VIP denotes the variable importance in projection value of the first principal component of the partial least squares-discriminant analysis (PLS-DA) model, which reflects the contribution of lipid compounds to group separation; FC represents the fold change, defined as the ratio of the mean quantitation values of all biological replicates for each lipid compound between the comparison groups. The screening thresholds were set as follows: $VIP > 1.0$, $FC > 1.2$ or $FC < 0.833$, and $P\text{-value} < 0.05$.

(C) The 20 lipid compounds with the largest fold changes in lipidomics analysis.

2. There appears to be another compound showing significant results (S1792). How do the authors explain why they did not include this compound in the manuscript? What exactly are those 23 FDA-approved lipid-regulating drugs? Providing an explanation or citing an appropriate reference would be useful.

Response: Thank you sincerely for your valuable advice and the insightful questions. At the initial stage of our research, we screened fluvastatin and S1792 through enzymatic experiments. Since S1792 exhibited weaker inhibitory activity against CYP4Z1 enzyme compared to fluvastatin, we did not include results regarding its effect on the stemness of breast cancer cells in manuscript. Specifically, we detected the impacts of fluvastatin and S1792 treatment on stemness of the MDA-MB-231 cell line: the proportion of stem cell subsets (CD24⁻CD44⁺) and the expression levels of stemness markers (ALDH1A1, OCT4, and SOX2). As presented in **Figure S1A-S1C**, fluvastatin showed a more potent inhibitory effect on breast cancer cell stemness than S1792, which led us to select fluvastatin as the focus of our study.

We also greatly appreciate you pointing out the lack of detailed information on FDA-approved lipid-regulating drugs. To address this, we have now supplemented the Catalog Number (Cat), official Name, and CAS Number of each drug in **Supplementary Table 1**.

Supplemental figure 1

The revised parts are shown below (red colored in the text). The corresponding passages are flagged with yellow background in the manuscript.

In **“Results”** section

Given that the lipid levels of breast cancer patients are abnormally upregulated during treatment and CYP4Z1 induces the upregulation of lipid metabolism in breast cancer cells, we screened potential anticancer drugs from a lipid-lowering compound library containing 23 FDA-approved drugs (**Supplementary Table 1**).

Specifically, we detected the impacts of fluvastatin and S1792 treatment on stemness of the MDA-MB-231 cell line. As presented in **Figure S1A-S1C**, fluvastatin showed a more potent inhibitory effect on breast cancer cell stemness than S1792.

In **“Discussion”** section

In our screening, simvastatin (S1792) was also found to exhibit inhibitory activity against CYP4Z1; regrettably, its pharmacological effect was less potent than that of fluvastatin (**Figure S1A-S1C**).

3. The design of the experiments in Figure 3 is confusing, with several pieces of information either missing or inconsistent.

– Is it necessary to use three different cell numbers for MDA-MB-231 cells? This is a

well-established cell line for in vivo experiments.

Response: Thank you for your valuable suggestions. We apologize for failing to provide an explanation of this part in the original manuscript. In our study, we initially planned to investigate the effects of fluvastatin on tumor formation, tumor growth, and stem cell proportion in breast cancer cells using the extreme limiting dilution assay [1]. Regrettably, the results did not align with our expectations—no significant difference was observed in the tumor formation rate among the three groups, which made it impossible to calculate the stem cell proportion. To ensure the authenticity and integrity of our research data, we decided to submit all experimental data (including the experiments with three different cell quantities) without omission.

[1] Hu, Y, and Smyth, GK (2009). ELDA: Extreme limiting dilution analysis for comparing depleted and enriched populations in stem cell and other assays. *Journal of Immunological Methods* 347, 70-78.

The revised parts are shown below (red colored in the text). The corresponding passages are flagged with yellow background in the manuscript.

In “**Results**” section

After subcutaneous xenograft tumors formed in Balb/c-nu nude mice, fluvastatin was administered at regular intervals of every seven days. The results demonstrated that fluvastatin effectively suppressed the growth of xenograft tumors, as evidenced by reduced tumor volume and/or weight compared to the control group. Regrettably, there was no significant difference in the number of tumors formed across groups with different cell inoculation amounts. Due to the lack of variability in tumor formation frequency, the ELDA could not be performed as originally planned (**Figure 3B-D**).

– Why was fluvastatin administered weekly in the tumor growth assay in Figure 3B, but every three days in Figure 3H?

Response: We sincerely apologize for the misunderstanding caused by our oversight. To clarify, Figure 3B corresponds to the ELDA model, which typically adopts a **low-frequency administration**. In contrast, Figure 3H depicts the intravenous-lung metastasis model, a system that requires **continuous administration**.

– What was the in vivo dose of fluvastatin? The figure legends are weak and do not provide clear explanations.

Response: Thank you for pointing out the missing information in our manuscript. To clarify, the dose of fluvastatin used in all *in vivo* experiments was **10 mg/kg**. We sincerely apologize for the oversight of not stating this key detail in manuscript. In response to your comment, we have now supplemented the detailed description in the figure legends throughout the entire manuscript to ensure clarity and completeness.

The revised parts are shown below (red colored in the text). The corresponding passages are flagged with yellow background in the manuscript.

In “**Results**” section

(B-D) Effects of fluvastatin administration on the size and weight of MDA-MB-231 and MCF-7-Adr xenograft tumors. Fluvastatin was administered at a dose of 10 mg/kg and dissolved in PBS.

– Why is the experimental window for fluvastatin administration so short? To assess potential toxicity, the drug should be administered for a longer duration (e.g., 4 weeks for primary tumor models).

Response: Thank you sincerely for your valuable suggestions, they have been instrumental in helping us refine the presentation of our study details. To address your concerns regarding the *in vivo* models in **Figure 3**, we first clarify that all models were established using human breast cancer cells, which is why we selected the BALB/c-nude mouse strain. To avoid potential immune recovery associated with mouse aging, we strictly used nude mice at 3 weeks of age and ensured the experiment concluded before the mice reached 8 weeks of age (the typical adult stage for this strain). Given the relatively long duration of drug administration in the intravenous-lung metastasis model, we considered that changes in mouse body weight could partially reflect the drug's potential toxic effects. We fully agree with your view that long-term continuous administration is essential for comprehensive toxicity assessment. Therefore, we further evaluated fluvastatin's *in vivo* toxicity using primary tumor models, and the corresponding results are presented in **Figure S7**.

– There is no body weight graph for the subcutaneous tumor model.

Response: Thank you for your attention to the details of our subcutaneous tumor model. As previously noted, in this specific model, the drug administration window was relatively short, and tumor volumes remained small throughout the experiment. Under such conditions, body weight changes are typically not sufficiently sensitive to reflect potential drug toxicity or the overall health status of the mice—this was the primary reason we did not include weight measurements in the current study. That said, we fully recognize that measuring body weight is a valuable and widely used approach for monitoring animal health and drug safety in preclinical studies. Your suggestion to incorporate this parameter is highly constructive, and we plan to include systematic body weight measurements in our subsequent research involving similar or extended *in vivo* models. This will help us more comprehensively evaluate the safety profile of experimental interventions and further enhance the rigor of our animal studies.

– In Figure 3F, are the tumors from the 1×10⁷ cell group or from another group? Why are there only three tumors in each group in Figure 3F, while Figure 3C shows six tumors in the graph?

Response: Thank you sincerely for your valuable suggestion, your attention to detail has helped us improve the rigor of our data analysis. To clarify, the tumors analyzed in **Figure 3F** were derived from the 1×10^7 cells group. In our initial analysis, we only selected the 3 largest tumors for evaluation, a choice that we now recognize was not optimal for ensuring the representativeness of the data. Prompted by your comment, we have revised this analysis: we re-evaluated all 6 tumors from this group to provide more comprehensive and statistically robust results. We sincerely apologize for this initial oversight and greatly appreciate you pointing it out.

4. Figure 4 could be strengthened by assessing changes in the PI3K/Akt pathway after exogenous addition of TAG, which would provide complementary evidence supporting the findings.

Response: Thank you for your constructive input. We fully concur with your insightful perspective. To further validate our findings and address your suggestion, we conducted additional experiments to examine the effects of exogenously added TAG and fluvastatin on the activation of the PI3K/Akt pathway. As presented in **Figure S4A**, the results show that fluvastatin inhibits the activation of the PI3K/Akt pathway, while exogenous TAG promotes its activation; notably, the supplementation of fluvastatin did not exert a significant rescue effect on the TAG-induced activation of this pathway.

The revised parts are shown below (red colored in the text). The corresponding passages are flagged with yellow background in the manuscript.

In “**Results**” section

After exogenous addition of TAG, stemness markers, p-PI3K and p-AKT were upregulated, and the down regulation effect of fluvastatin administration on them was negligible (**Figure 4B and S4A**).

5. Some parts of the manuscript give the impression of being drafted quickly, and a more careful revision could greatly improve clarity and readability.

Response: Thank you sincerely for your valuable suggestions. We greatly appreciate your meticulous review, which has helped us identify areas for improvement in manuscript. We conducted a comprehensive check and careful polishing of the entire text to address any potential ambiguities and enhance the overall rigor and readability of the manuscript.

6. Please specify the duration of in vitro treatment with fluvastatin; Please clearly indicate the solvent used for fluvastatin, either in the main text or in the Methods section.

Response: Thanks for your advice, which contributes significantly to enhancing the rigor of our work. We have carefully addressed your feedback by supplementing the relevant details in both “**Materials and Methods**” and “**Figure legends**” section.

The revised parts are shown below (red colored in the text). The corresponding passages are flagged with yellow background in the manuscript.

In “Materials and methods” section

Cell culture and reagents

Cells were treated with all drugs for 48 h in vitro. The solvent of fluvastatin was PBS, and the solvent of TAG and A922500 was DMSO.

Reviewer #2

1. In both the *in vitro* and *in vivo* models, only CYP4Z1 overexpression systems were used. Because CYP4Z1 overexpression represents an artificial condition, knockdown or knockout models are needed to demonstrate that CYP4Z1 is critical for the anti-tumor effects of fluvastatin.

Response: Thanks for your constructive input, which contributes significantly to enhancing the rigor of our work. Regarding the *in vivo* model selection, it is worth noting that CYP4Z1 expression has, to date, only been identified in primates (PMID: 15059886). Given that our research is still in the preliminary exploration stage and constrained by limited funding, we were unable to conduct *in vivo* experiments using primate models. Instead, we generated KI-CYP4Z1 C57BL/6 mice, and as shown in **Figure 6**, CYP4Z1 knock-in significantly promoted tumor progression and metastasis in the primary tumor model. Notably, while fluvastatin exhibited therapeutic effects in mice without CYP4Z1 expression, its efficacy was far more pronounced in the KI-CYP4Z1 model—specifically, it almost completely rescued the tumor progression and metastasis induced by CYP4Z1 knock-in. This observation partially demonstrates the targeting of CYP4Z1 by fluvastatin.

We fully agree with your view that a CYP4Z1 knockdown model is essential to validate our findings. Therefore, we established *in vitro* CYP4Z1 knockdown models using human breast cancer cell lines (MDA-MB-231 and MCF-7-Adr) and examined cell stemness, PI3K/AKT pathway activation, migration, and invasion. The Western blotting results showed that both fluvastatin treatment and CYP4Z1 knockdown attenuated breast cancer cell stemness and inhibited PI3K/AKT pathway activation (**Figure S4H**). Sphere formation assays revealed that fluvastatin treatment and CYP4Z1 knockdown reduced the proportion of cancer stem cells in breast cancer cells (**Figure S4M-S4N**). Additionally, Transwell assays demonstrated that both interventions impaired the migration and invasion capabilities of breast cancer cells (**Figure S4I-S4L**). To investigate whether there is a difference in the efficacy of fluvastatin between CYP4Z1 knockdown and normal models, we compared the relative inhibition rates of the aforementioned indicators. The results showed that in the CYP4Z1 knockdown model, the relative inhibition rates of fluvastatin on most indicators were significantly lower than those in the normal model—this finding indicates that CYP4Z1 is a key target of fluvastatin (**Figure S4O-S4P**). However, fluvastatin still exerted a certain degree of efficacy in the CYP4Z1 knockdown model, which suggests that fluvastatin may act on additional targets beyond CYP4Z1 (**Figure S4O-S4P**). We hope this explanation thoroughly addresses your concerns. Once again, we are grateful for

your meticulous review and insightful feedback, which have been crucial for deepening the mechanistic exploration of our study.

The revised parts are shown below (red colored in the text). The corresponding passages are flagged with yellow background in the manuscript.

In “**Materials and methods**” section

Plasmid, si-CYP4Z1 and cell transfection

si-CYP4Z1 sequence: -CAUUACCUUCCAGAUGGATTdTdT-

In “**Results**” section

Furthermore, we established *in vitro* CYP4Z1 knockdown models. Both fluvastatin treatment and CYP4Z1 knockdown attenuated breast cancer cell stemness and inhibited PI3K/AKT pathway activation (**Figure S4H**). Consistent with these findings, both interventions reduced the proportion of cancer stem cells in breast cancer cells (**Figure S4M-S4N**). Additionally, Transwell assays further confirmed that these two treatments impaired the migration and invasion capabilities of breast cancer cells (**Figure S4I-S4L**). To investigate whether there is a difference in the efficacy of fluvastatin between CYP4Z1 knockdown and normal models, we compared the relative inhibition rates of the aforementioned indicators. The results showed that in the CYP4Z1

knockdown model, the relative inhibition rates of fluvastatin on most indicators were significantly lower than those in the normal model—this finding indicates that CYP4Z1 is a key target of fluvastatin (Figure S4O-S4P). However, fluvastatin still exerted a certain degree of efficacy in the CYP4Z1 knockdown model, which suggests that fluvastatin may act on additional targets beyond CYP4Z1 (Figure S4O-S4P).

In “Discussion” section

We also discovered that CYP4Z1 knockdown models and PyMT-MMTV mice responded to fluvastatin (Figure 6 and S4H-S4N). These results suggest that fluvastatin has multiple roles in the occurrence and development of breast cancer, and there may be other potential targets, which deserve further investigation. Nevertheless, in both in vitro and in vivo models, the efficacy of fluvastatin in models expressing CYP4Z1 was significantly superior to that in models not expressing CYP4Z1 (Figure S4O-S4P and S7D). Therefore, there is no denying that CYP4Z1 is one of the key targets through which fluvastatin exerts its anti-tumor effects.

2. In Figure 4A, CYP4Z1 overexpression does not appear to markedly affect PI3K/AKT phosphorylation, contrary to the authors’ claim.

Response: Thank you for your attention to the details of the Western blotting results. We would like to clarify the observations regarding the p-PI3K/PI3K and p-AKT/AKT ratio changes across different lanes, as well as the follow-up validation we conducted. For the initial WB data, the changes in p-PI3K/PI3K and p-AKT/AKT ratios in Lanes 1 and 3 were not obvious, but the comparison between Lanes 2 and 4 clearly showed that CYP4Z1 overexpression significantly increased the p-PI3K/PI3K and p-AKT/AKT ratios, which is consistent with our expected experimental outcomes. To further confirm the reliability of our conclusions, we replicated WB experiments. The results verified our original expectation, with the p-PI3K/PI3K and p-AKT/AKT ratio changes induced by CYP4Z1 overexpression being clearly and reproducibly observed.

3. In Figure 5, the results appear contradictory. On one hand, all of the mutants exhibited reduced enzymatic activity (Fig. 5C); on the other hand, overexpression of these mutants led to increased stem cell marker expression (Fig. 5E). It remains unclear whether these mutants also activate downstream signaling pathways, such as PI3K/AKT.

Response: Thank you for your meticulous review and thoughtful attention to these details—your feedback has been invaluable in refining the clarity of our experimental logic and conclusions. We would like to clarify the details of the CYP4Z1 mutations experiment to address potential ambiguities. Specifically, we only introduced **single-amino-acid mutations** into CYP4Z1, and these mutations led to two key outcomes: a significant reduction in the enzyme’s activity (Figure 5C) and a loss of its binding ability to fluvastatin (Figure 5D). These results strongly suggest that the three amino acid sites are most likely located in the **active center** of CYP4Z1, and that the

binding of fluvastatin to CYP4Z1 is directly influenced by these sites. Regarding the residual activity of the mutants: as shown in **Figure 5C**, compared with the negative control, the CYP4Z1 mutants still retained partial enzyme activity. This residual activity provides a reasonable mechanistic basis for the observation that the mutants could still upregulate the expression of stemness markers—since even reduced enzymatic activity may be sufficient to modulate downstream processes related to cell stemness.

We also shared your concern about whether the mutants would affect the activation of downstream signaling pathways. To verify this, we conducted additional experiments, and the results were consistent with the changes in enzyme activity and stemness markers: both wild-type CYP4Z1 and the mutants promoted the activation of the PI3K/AKT pathway, but the magnitude of pathway activation induced by the mutant was significantly weaker than that induced by wild-type CYP4Z1 (**Figure S6A**). This further confirms that the enzymatic activity of CYP4Z1 is closely linked to its ability to regulate the PI3K/AKT pathway.

Supplemental figure 6

The revised parts are shown below (red colored in the text). The corresponding passages are flagged with yellow background in the manuscript.

In “Results” section

Both the wild-type CYP4Z1 and mutants were found to enhance cell stemness and promote the activation of the PI3K/AKT pathway. However, a key distinction emerged in the magnitude of these effects: the extent of stemness enhancement and PI3K/AKT pathway activation induced by each of the three mutants was weaker than that induced by the wild-type CYP4Z1 (**Figure 5E and S6A**).

4. In Figures 6D–E, the quality of the FACS data is poor. Furthermore, while the CD24⁺CD29^{hi} population is known to be enriched for stem/progenitor cells, the CD24⁺CD29^{lo} population also retains some progenitor cell properties. Overexpression of CYP4Z1 is expected to expand the CD24⁺CD29^{hi} population. In addition, previous publications report that the CD24⁺CD29^{lo} population accounts for approximately 20–30% of mammary epithelial cells.

Response: Thank you again for your meticulous review and insightful feedback. We fully agree with your valuable perspective. With your reminder, we realized there was an error in our previous gating strategy. Therefore, we **adjusted the gating strategy** for the FACS data. The updated results revealed two key observations (**Figure 6D–E**): (1) For the CD24⁺ CD29^{lo} cell subset: KI-CYP4Z1 led to an increased proportion of this subset, but fluvastatin treatment had no significant effect on it. (2) For the CD24⁺ CD29^{hi} cell subset: KI-CYP4Z1 also elevated the proportion of this subset, and notably, fluvastatin treatment significantly reduced its proportion.

Additionally, the observation that fluvastatin exhibited therapeutic effects in the wild-type spontaneous tumor model suggests that fluvastatin may act on other targets beyond CYP4Z1. However, the critical finding that fluvastatin treatment completely abrogated the KI-CYP4Z1-induced increase in cancer stem cell subsets clearly indicates that CYP4Z1 is one of the key targets through which fluvastatin exerts its anti-tumor effects.

We hope this adjustment to the FACS gating strategy and the corresponding result interpretation address your concerns.

The revised parts are shown below (red colored in the text). The corresponding passages are flagged with yellow background in the manuscript.

In “Results” section

Given that the development of breast cancer in mice is associated with mammary luminal progenitor cells and stem cells, and considering that the CD24⁺CD29^{hi} population is known to be enriched for stem/progenitor cells while the CD24⁺CD29^{lo} population retains some progenitor properties, we performed primary isolation and sorting of mammary tissues. Flow cytometry analysis revealed that knock in *CYP4Z1* increased the proportion of the CD24⁺CD29^{lo} cell subset in mammary epithelial cells of *PyMT-MMTV* mice, with no effect of fluvastatin administration on this subset. Meanwhile, knock in *CYP4Z1* also elevated the proportion of the CD24⁺CD29^{hi} subset, and fluvastatin treatment significantly reduced the proportion of this specific subset (**Figure 6D-E**).

5. The legends are too short and lacks some details.

Response: Thanks for your advice, which contributes significantly to enhancing the rigor of our work. We have carefully addressed your feedback by supplementing the relevant details in “**Figure legends**” section.

Reviewer #3

1. Fluvastatin also affected PyMT-MMTV mice without CYP4Z1, suggesting off-target effects needing clarification. Your study shows fluvastatin suppresses breast cancer stem cells via CYP4Z1. However, PyMT-MMTV control mice without CYP4Z1 also responded, implying additional mechanisms. What other targets or pathways might explain this effect? Have you considered its role as an HMG-CoA reductase inhibitor and potential CYP4Z1-independent impact on breast cancer?

Response: Thank you for your constructive input. We fully concur with your insightful perspective. To further investigate whether fluvastatin exerts off-target effects, we established *in vitro* CYP4Z1 knockdown models using human breast cancer cell lines (MDA-MB-231 and MCF-7-Adr), and assessed cell stemness, PI3K/AKT pathway activation, migration, and invasion (**Figure S4H-S4N**).

To investigate whether there is a difference in the efficacy of fluvastatin between CYP4Z1 knockdown and normal models, we compared the relative inhibition rates of the aforementioned indicators. Consistent with the *in vivo* model results (**Figure S7D**), in the CYP4Z1 knockdown model, the relative inhibition rates effects of fluvastatin on most indicators were significantly lower than those in the normal model—this finding indicates that CYP4Z1 is a key target of fluvastatin (**Figure S4O-S4P**). However, fluvastatin still exerted a certain degree of efficacy in the CYP4Z1 knockdown model, which suggests that fluvastatin may act on additional targets beyond CYP4Z1 (**Figure S4O-S4P**).

A large body of existing research has demonstrated that statins (HMG-CoA reductase inhibitors) possess anti-tumor activity (PMID: 36650022, 39143707, 34627266, 32694178, 38816352),

which was one of the key considerations in our selection of drug library. Notably, compared with simvastatin (S1792, which also exhibits CYP4Z1 inhibitory activity), fluvastatin showed more potent activity in suppressing cancer cell stemness (**Figure S1A-S1C**)—a finding indicating that CYP4Z1 enzymatic activity is one of the important drivers of breast cancer cell stemness.

Supplemental figure 1

Additionally, our data showed that CYP4Z1 promotes the biosynthesis of TAG (**Figure 1B-C**), which is consistent with the clinically observed phenomenon of abnormal hyperlipidemia in breast cancer patients. As an HMG-CoA reductase inhibitor, fluvastatin can reduce TAG levels to alleviate hyperlipidemia. Therefore, we propose that fluvastatin exerts its anti-tumor effects through two complementary mechanisms: first, it directly targets CYP4Z1 to inhibit its enzymatic activity; second, it indirectly reduces TAG levels via its HMG-CoA reductase inhibitory activity. This HMG-CoA reductase-mediated activity represents an important CYP4Z1-independent anti-tumor pathway of fluvastatin.

We hope this explanation thoroughly elaborates on the multi-target and multi-pathway mechanisms of fluvastatin. Thank you again for your meticulous review and thoughtful feedback, which have been instrumental in deepening the mechanistic discussion of our study.

The revised parts are shown below (red colored in the text). The corresponding passages are flagged with yellow background in the manuscript.

In **“Materials and methods”** section

Plasmid, si-CYP4Z1 and cell transfection

si-CYP4Z1 sequence: -CAUUACCUUCCAGAUGGATTdTdT-

In **“Results”** section

Specifically, we detected the impacts of fluvastatin and S1792 treatment on stemness of the MDA-MB-231 cell line: the proportion of stem cell subsets (CD24⁺CD44⁺) and the expression levels of stemness markers (ALDH1A1, OCT4, and SOX2). As presented in **Figure S1A-S1C**, fluvastatin showed a more potent inhibitory effect on breast cancer cell stemness than S1792.

Furthermore, we established in vitro CYP4Z1 knockdown models. Both fluvastatin treatment and CYP4Z1 knockdown attenuated breast cancer cell stemness and inhibited PI3K/AKT pathway activation (**Figure S4H**). Consistent with these findings, both interventions reduced the proportion of cancer stem cells in breast cancer cells (**Figure S4M-S4N**). Additionally, Transwell assays further confirmed that these two treatments impaired the migration and invasion capabilities of breast cancer cells (**Figure S4I-S4L**). To investigate whether there is a difference in the efficacy of fluvastatin between CYP4Z1 knockdown and normal models, we compared the relative

inhibition rates of the aforementioned indicators. The results showed that in the CYP4Z1 knockdown model, the relative inhibition rates of fluvastatin on most indicators were significantly lower than those in the normal model—this finding indicates that CYP4Z1 is a key target of fluvastatin (**Figure S4O-S4P**). However, fluvastatin still exerted a certain degree of efficacy in the CYP4Z1 knockdown model, which suggests that fluvastatin may act on additional targets beyond CYP4Z1 (**Figure S4O-S4P**).

In “**Discussion**” section

In our screening, simvastatin (S1792) was also found to exhibit inhibitory activity against CYP4Z1; regrettably, its pharmacological effect was less potent than that of fluvastatin (**Figure S1A-S1C**).

We also discovered that CYP4Z1 knockdown models and *PyMT-MMTV* mice responded to fluvastatin (**Figure 6 and S4H-S4N**). These results suggest that fluvastatin has multiple roles in the occurrence and development of breast cancer, and there may be other potential targets, which deserve further investigation. Nevertheless, in both in vitro and in vivo models, the efficacy of fluvastatin in models expressing CYP4Z1 was significantly superior to that in models not expressing CYP4Z1 (**Figure S4O-S4P and S7D**). Therefore, there is no denying that CYP4Z1 is one of the key targets through which fluvastatin exerts its anti-tumor effects.

2. CYP4Z1 overexpression elevates lipid metabolism and TAG levels in breast cancer cells, linking it to lipid peroxidation—the key driver of ferroptosis. Could fluvastatin's inhibitory effect on CYP4Z1 and lipid metabolism be a mechanism for inducing ferroptosis in breast cancer cells? Have you considered measuring markers of ferroptosis, such as lipid peroxidation (e.g., malondialdehyde or 4-hydroxynonenal) or the expression levels of key regulators like GPX4 and ACSL4, in your in vitro or in vivo models? Exploring this pathway would provide a more complete picture of how fluvastatin's modulation of lipid metabolism leads to tumor suppression.

Response: Thank you for your visionary suggestion—we highly agree with your hypothesis and therefore conducted experiments to investigate whether fluvastatin induces ferroptosis in breast cancer cells.

To address this question, we designed a series of targeted assays focusing on key molecular and histological markers of ferroptosis. First, we measured the intracellular glutathione (GSH) content—a critical antioxidant that regulates ferroptosis sensitivity. The results showed that neither CYP4Z1 overexpression nor fluvastatin treatment had a significant impact on GSH levels (**Figure S8A**). Next, we examined the expression levels of GPX4 (a core enzyme that suppresses ferroptosis by reducing lipid peroxides) and ACSL4 (a key mediator that promotes ferroptosis by regulating lipid metabolism). Regrettably, no significant changes in the expression of these two markers were observed across experimental groups (**Figure S8B**). Finally, we performed Prussian blue DAB-enhanced staining on tumor tissues to assess intratumoral iron deposition (a hallmark of ferroptosis). This histological analysis revealed that fluvastatin treatment did not alter iron accumulation within the tumor (**Figure S8C**). Collectively, these experimental results consistently indicate that fluvastatin does not induce ferroptosis in breast cancer cells.

Supplemental figure 8

We greatly appreciate your insightful hypothesis, which guided us to explore this important potential mechanism and further enrich the understanding of fluvastatin’s anti-tumor effects. Thank you again for your meticulous review and constructive feedback, which help strengthen the comprehensiveness of our study.

The revised parts are shown below (red colored in the text). The corresponding passages are flagged with yellow background in the manuscript.

In “**Materials and methods**” section

Intracellular Glutathione (GSH) Detection Assay

Collect cells (5×10^5), wash twice with PBS to remove extracellular GSH. Lyse cells if needed, then centrifuge lysates at high speed for 10 min. Add 50 μ L cell lysate or GSH standard to a microcentrifuge tube, followed by 150 μ L 10% TCA. Vortex, incubate on ice for 10 min, then centrifuge at high speed for 10 min. Prepare 0.4 M EDTA solution. Transfer 50 μ L supernatant to a new tube, add 200 μ L 0.4 M EDTA and 25 μ L 10 mM DTNB. Vortex, incubate at room temperature for 10 min. Measure absorbance at 412 nm with a spectrophotometer (transfer 100 μ L mixture to a 96-well plate for analysis if needed). Calculate GSH concentration using a GSH standard curve (Abbkine, KTB1600-48T, China).

Prussian blue DAB-enhanced staining

Prussian blue DAB-enhanced staining was performed by Shanghai Ruchuang Biological Technology Co., Ltd., following the protocol: Paraffin blocks were sectioned, and the resulting sections were baked at 60°C. Subsequently, sections were dewaxed and dehydrated using xylene and absolute ethanol. Perls A and Perls B solutions were mixed at a 1:1 ratio, and sections were incubated in this mixture for 30 min for Prussian blue staining. Afterward, DAB chromogenic solution was dropped onto the sections for 2 min to develop color. Sections were then stained with hematoxylin for 30 s. Finally, the stained sections were dehydrated and mounted with neutral balsam.

In “**Results**” section

Notably, even though fluvastatin exhibited superior efficacy in models expressing CYP4Z1 compared to those not expressing CYP4Z1, both the CYP4Z1 knockdown models and the PyMT-MMTV model still responded to fluvastatin (**Figure S4O-S4P and S7D**)—this suggests that fluvastatin possesses additional anti-tumor targets beyond CYP4Z1. Our results demonstrated that CYP4Z1 overexpression enhanced lipid metabolism and increased TAG levels in breast cancer cells (**Figure 1B-C**). This observation raises the questions: given that lipid peroxidation is a critical driver of ferroptosis, does CYP4Z1 associate with ferroptosis? And could the activity of

fluvastatin in regulating lipid metabolism be one of the mechanisms underlying its induction of ferroptosis in breast cancer cells? To address this question, we measured the intracellular GSH content, the results showed that neither CYP4Z1 overexpression nor fluvastatin treatment had a significant impact on GSH levels (**Figure S7A**). Additionally, we examined the expression levels of GPX4 and ACSL4. Regrettably, no significant changes in the expression of these two markers were observed across experimental groups (**Figure S7B**). Consistently, Prussian blue DAB-enhanced staining analysis revealed that fluvastatin treatment did not alter iron accumulation within the tumor (**Figure S7C**). Collectively, these results consistently indicate that fluvastatin does not induce ferroptosis in breast cancer cells.

In “**Discussion**” section

These results suggest that fluvastatin has multiple roles in the occurrence and development of breast cancer, and there may be other potential targets (excluding ferroptosis, **Figure S7**), which deserve further investigation.

We are very glad to get helpful suggestions from the editor and reviewers for revision. Thank you so much. The manuscript has been resubmitted to your journal. We are looking forward to your positive response.

Best regards

Lufeng Zheng

China Pharmaceutical University, Nanjing, 210009 China